# Unified Neural Network Scaling Laws and Scale-time Equivalence

## Abstract

As neural networks continue to grow in size but datasets might not, it is vital to understand how much performance improvement can be expected: is it more important to scale network size or data volume? Thus, neural network scaling laws, which characterize how test error varies with network size and data volume, have become increasingly important. However, existing scaling laws are often applicable only in limited regimes and often do not incorporate or predict well-known phenomena such as double descent. Here, we present a novel theoretical characterization of how *three* factors — model size, training time, and data volume — interact to determine the performance of deep neural networks. We first establish a theoretical and empirical equivalence between scaling the size of a neural network and increasing its training time proportionally. Scale-time equivalence challenges the current practice, wherein large models are trained for small durations, and suggests that smaller models trained over extended periods could match their efficacy. It also leads to a novel method for predicting the performance of large-scale networks from small-scale networks trained for extended epochs, and vice versa. We next combine scale-time equivalence with a linear model analysis of double descent to obtain a unified theoretical scaling law, which we confirm with experiments across vision benchmarks and network architectures. These laws explain several previously unexplained phenomena: reduced data requirements for generalization in larger models, heightened sensitivity to label noise in overparameterized models, and instances where increasing model scale does not necessarily enhance performance. Our findings hold significant implications for the practical deployment of neural networks, offering a more accessible and efficient path to training and fine-tuning large models.

## 1 Introduction

Progress in artificial intelligence (AI) has relied heavily on the dramatic growth in the size of models and datasets. An active area of research focuses on understanding how test error decreases with increases in model and data size. This work has led to the development of scaling laws which posit that test error decreases as a power law with both. However, several theoretical aspects remain unclear. One significant gap is understanding how test error and the existing scaling laws change as the training time is varied (Kaplan et al., 2020; Bahri et al., 2021; Rosenfeld et al., 2020; Sharma & Kaplan, 2022).

The practical relevance of this question is clear: under a fixed compute budget, what is the optimal balance between scaling the model size and dataset volume, and what is the right amount of training for a given data volume? This is particularly relevant in the context of large language models (LLMs), which are often trained for a single epoch, raising questions about the potential efficacy of training smaller models for longer (more epochs).

Furthermore, current scaling laws do not account for other well-known phenomena in learning, such as double descent (Belkin et al., 2019), in which model performance exhibits non-monotonic changes with respect to training data volume, model size, and training time. In particular, double descent theory predicts that test error should increase rapidly at the *interpolation threshold*, the point at which the model interpolates the training set (Nakkiran et al., 2021; Advani & Saxe, 2017). Like scaling laws, current theories of double descent leave several empirical phenomena unexplained:

past explanations of double descent require it to occur, but empirically double descent is often not observed; it is unclear whether the interpolation threshold should grow or shrink with model size; prior theory does not explain why models in the infinite-parameter limit sometimes perform worse than their finite-parameter counterparts.

We seek the simplest possible unified framework in which to understand learning with respect to model size, data volume, and training time. In doing so, we aim to capture the essential scaling properties of learning in deep neural networks. Our results unify double descent with scaling laws and help to understand when models are sufficiently large for effective performance, how double descent is affected by varying training time and model size, and the variability and shape of loss curves across different problems.

The contributions of this paper are multi-fold:

- We theoretically and empirically demonstrate that scaling the size of a neural network is functionally equivalent to increasing its training time by a proportional factor.

- Leveraging this insight, we 1) predict the performance of large-scale networks using small-scale networks trained for many epochs and 2) predict the performance of networks trained for many epochs using the performance of large networks trained for one epoch.

- Using scale-time equivalence, we propose a unified scaling law for deep neural networks that provides a new explanation for parameter-wise double descent: double descent occurs when small models, which effectively train slower than larger models, acquire noisy data features.

- Through experiments conducted on standard vision benchmarks across multiple network architectures, we validate that our model explains several previously unexplained phenomena, including 1) the reduced data requirement for generalization in larger models, 2) the large impact of label noise on overparameterized models, 3) why error of overparameterized models often *increases* with scale.

## 2 RELATED WORK

### 2.1 SCALING LAWS

Neural network scaling laws describe how generalization error scales with data and model size. A number of works have observed power-law scaling with respect to data and model size (Kaplan et al., 2020; Rosenfeld et al., 2020; Clark et al., 2022), which has been explained theoretically (Bahri et al., 2021; Sharma & Kaplan, 2022; Hutter, 2021; Paquette et al., 2024).

However, other work demonstrates that model scale may not be sufficient to predict model performance (Tay et al., 2022), and casts doubt on power laws as the best model of error rate scaling (Alabdulmohsin et al., 2022; Bansal et al., 2022; Mahmood et al., 2022). Moreover, scaling with respect to training time, holding data volume fixed, remains poorly understood. These observations highlight the need for a more general framework that can predict model performance under many settings.

### 2.2 DOUBLE DESCENT

Double descent is an empirically observed phenomenon in which generalization error of machine learning models with respect to training data volume, model size, and training time exhibits an initial decrease, followed by a brief, sharp increase followed by a final decrease (Belkin et al., 2019; Nakkiran et al., 2021). Double descent with respect to model and data size has been theoretically understood as occurring due to a high degree of overfitting at the *interpolation threshold*, the point at which the model size is just sufficient to interpolate the training data (Adlam & Pennington, 2020; D'Ascoli et al., 2020; Belkin et al., 2019; Advani & Saxe, 2017). Typically, this work uses tools from random matrix theory to explain double descent for random feature models, in which linear regression maps a random, fixed feature pool to the desired output (Simon et al., 2024; Atanasov et al., 2024; Adlam et al., 2022; Bordelon et al., 2024; Maloney et al., 2022; Mei & Montanari, 2019; Ali et al., 2019; Lin et al., 2024).

However, these models generally do not explain double descent behavior in terms of training time. Originally, Nakkiran et al. (2019; 2021) hypothesized that training models for longer results in an effective increase in model size, thus allowing time-wise double descent to be explained in the same way as scale-wise double descent. Recent works have provided an alternate explanation: time-wise double descent occurs due to data features being learned at varying scales (Pezeshki et al., 2022; Heckel & Yilmaz, 2021; Stephenson & Lee, 2021). Error rises as models overfit to quickly-learned noisy features, but then falls as models more slowly learn signal features. In this work, we unify this explanation of time-wise double descent with the traditional account of double descent.

# 3 SCALE-TIME EQUIVALENCE IN NEURAL NETWORKS

In this section, we demonstrate that model size and training time may be traded off with each other. This result is consistent with and generalizes prior results demonstrating that models learn functions of increasing complexity over time (Nakkiran et al., 2019). We establish the result theoretically in a simplified model and validate it empirically in neural networks across several datasets and architectures.

## 3.1 RANDOM SUBSPACE MODEL

Following the lines of prior double descent analyses in random feature models, we construct a random subspace model to demonstrate scale-time equivalence theoretically. Consider a large $P$ dimensional model with parameters $\beta$, such that the function represented by the model depends only on a low-dimensional linear projection of $\beta$. This is reasonable for neural networks: it is well known that neural networks trained by stochastic gradient descent tend towards flat minima of their loss landscapes, thus revealing many redundant dimensions in the network (i.e. only a low-dimensional subspace affects the network output). Specifically, we denote the low-dimensional projection as $\alpha \in \mathbb{R}^r$ where $r < P$ which is constructed as:

$$\alpha = K\beta \tag{1}$$

where $K \in \mathbb{R}^{r \times P}$ is a fixed projection matrix.

Assume that we can only control a random $p$-dimensional ($P > p > r$) linear subspace of the large model. This may again be a reasonable assumption for model classes such as neural networks: smaller neural networks can naturally be viewed as linear subspaces of larger neural networks that contain them architecturally (i.e. have a larger width and depth) (Hall & Li, 1993). We denote the parameters of our controllable $p$ dimensional subspace model as $\theta \in \mathbb{R}^p$, where:

$$\beta = R\theta + \beta_0 \tag{2}$$

for a random matrix $R \in \mathbb{R}^{P \times p}$ with elements drawn iid from a unit Gaussian and with $\beta_0 \in \mathbb{R}^P$ fixed. We will represent time $t$ with subscript $t$ (i.e. $\beta_0$ denotes $\beta$ at time 0). Observe that any $p$ dimensional affine subspace of $\mathbb{R}^P$ may be represented in the form above for some choice of $R$ and $\beta_0$.

Under these assumptions, we can show that under gradient descent, increasing the scale $p$ of the model is equivalent to increasing training time:

**Theorem 1.** *We denote the loss as a function of $\alpha$: $L \in \mathbb{R}^r \to \mathbb{R}$. Suppose $L$ has Lipschitz constant $l$ and its second derivative has Lipschitz constant $h$. Suppose that continuous time gradient flow is applied to $\theta$ with learning rate $\eta$ from initialization $\theta = 0$. We denote $\alpha_t = K(R\theta_t + \beta_0)$ where $\theta_t$ are the parameters at time $t$. Denote $A_t \in \mathbb{R}^r$ as the solution to:*

$$\dot{A}_t = -\eta K K^T \nabla L(A_t) \tag{3}$$

*with initial condition $A_0 = K\beta_0$. Note that $A_t$ does not depend on $p$. Then, with probability $1 - \epsilon$:*

$$||\alpha_t - A_{pt}|| \leq \frac{l\sqrt{||K||_F^4 + ||KK^T||_F^2}}{h\sqrt{p\epsilon}||KK^T||}(e^{\eta pth||KK^T||} - 1) \tag{4}$$

See Appendix A for a proof. The theorem implies closeness between the function implemented by the network, represented by $\alpha_t$, and another quantity $A_{pt}$ which *only depends on the product $pt$*. In

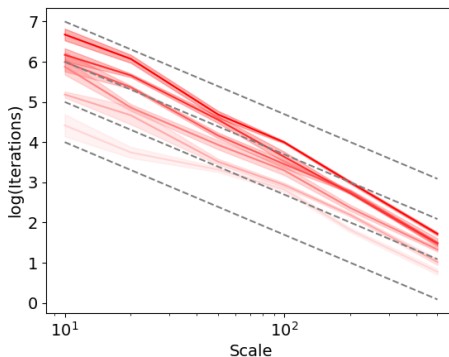

Figure 1: **Proportional trade-off between model scale and training time: testing the prediction on a linear model.** Red lines indicate tradeoff curves between number of training iterations and model size. Curves are computed by, for each model size, measuring the minimum amount of training time necessary to achieve different loss levels. Different curves indicate different performance thresholds; darker lines indicate a smaller error threshold. Margins indicate standard errors over 5 trials. Grey dashed lines represent 1:1 proportionality between scale and training iterations.

other words, the learned model only depends on the product of the number of parameters $p$ and time $t$ (up to some error). Thus, we may interpret the product $pt$ as representing the distance of a model along the training trajectory; larger $pt$ implies more training progress. Increasing the number of parameters by a scale factor is equivalent to increasing the training time by the same scale factor and vice versa: scale is *equivalent* to time. Intuitively, this is because each parameter allows the function to learn at a fixed rate; thus, adding more parameters linearly increases the effective learning rate.

The result also reveals when such a scale-time equivalence *cannot* be made. The error bound implies that when training progress $pt$ is fixed, as $p$ grows, scale and time become increasingly equivalent (the bound approaches zero). The equivalence breaks down for small $p$ since here, the randomly chosen subspace of the model may or may not align well with $K$; as $p$ grows, the amount of alignment becomes less stochastic. Moreover, as training progress $pt$ grows, the bound grows exponentially because small perturbations to the model early in the training trajectory lead to exponentially larger changes later in the trajectory. Finally, we highlight that our result holds under standard neural network parameterizations in which the gradient of the model output with respect to each parameter does not scale with $p$; in other parameterizations such as Neural Tangent Kernel (Jacot et al., 2018), we may expect a different form of scale-time equivalence.

We first validate our prediction in a simple linear model in which a varying fraction of the model parameters are controllable. See Appendix C for details. In Figure 1, we find lines of 1:1 proportionality between model scale and the number of iterations required to reach a fixed loss level, validating our theory.

## 3.2 EMPIRICAL VALIDATION IN NEURAL NETWORKS

We next turn to examine whether scale-time equivalence is present empirically in neural networks. We conduct experiments on MNIST (Deng, 2012), CIFAR-10 (Krizhevsky, 2009), and SVHN (Goodfellow et al., 2013) training a 7-layer convolutional neural network (CNN) and a 6-layer multilayer perception (MLP) with stochastic gradient descent (SGD). To assess scale-time equivalence, we measure the *minimum* amount of training time required to achieve non-zero generalization under various network widths and by varying the dataset size by subsampling. Scale-time equivalence predicts that wider networks will require less time to generalize in a systematically predictable way. See Appendix C for further experimental details.

As observed in Figure 2, in all settings, we see a clear tradeoff curve between scale and training time: increasing scale by a fixed factor is nearly equivalent to reducing training time by another fixed factor. Importantly, the scale here is set as the *effective* number of network parameters, defined as the maximum number of training points that can be fit by the network, not the absolute number

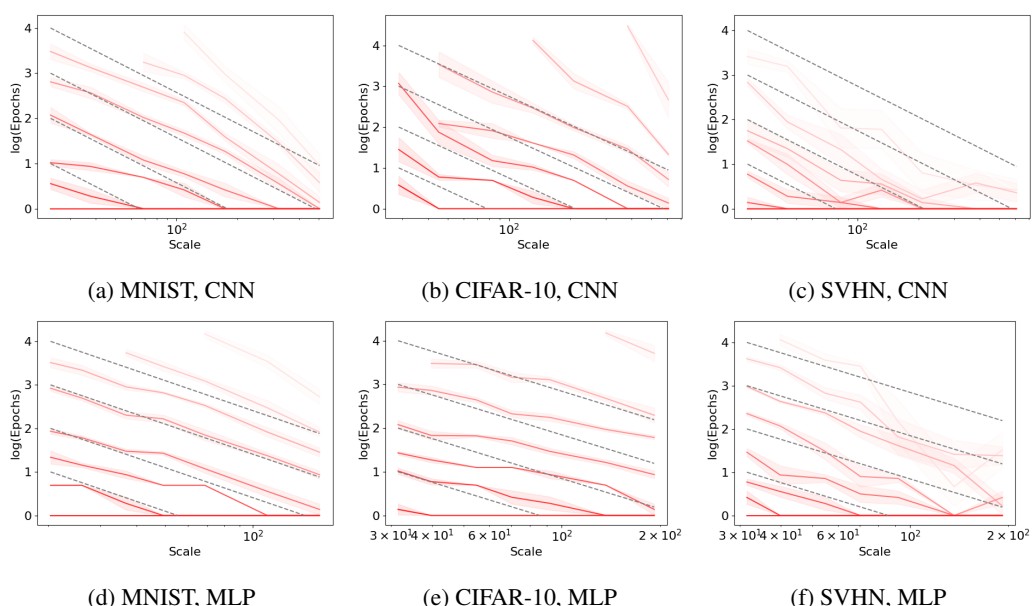

Figure 2: **Proportional trade-off between model scale and training time: testing the prediction on neural networks.** Red lines indicate tradeoff curves between number of training epochs and network scale for different datasets and architectures trained with SGD. Different curves indicate different amounts of training data; darker lines indicate more data. Curves are computed by, for each network scale, measuring the minimum amount of training time necessary to achieve non-zero generalization. Margins indicate standard errors over 5 trials. Grey curves are lines of 1:1 proportionality between scale and training epochs.

of parameters. We set the effective parameter count as the *cube root* of the number of parameters. Appendix B provides a heuristic argument for this scaling rate.

Under this choice of scale, we find a systematic and predictable relationship between scale and training time, demonstrating that scale-time equivalence can be empirically observed. We also note that this phenomenon has been observed in prior literature (although not quantitatively characterized); for instance, Nakkiran et al. (2019) find that patterns of double descent are similar with respect to training epochs and network size. We emphasize that these results are limited to gradient descent (the setting of our theory). With Adam optimization (Kingma & Ba, 2015), the number of epochs for generalization first decreases with scale, then increases (see Appendix D Figure 7). We hypothesize that since Adam has an adaptive learning rate, it is using a smaller effective learning rate for very large networks. As learning rate shrinks, more epochs are needed to generalize, leading to the observed results.

## 4 PREDICTING OPTIMAL NETWORK SCALE AND TRAINING TIME

Scale-time equivalence suggests it should be possible to predict the performance of large models by training small models for many epochs and vice versa. This allows us to predict the optimal network scale and training time for a given dataset and base architecture.

On benchmark datasets and architectures, we conduct two experiments: 1) predict performance under varying model scales from a small network trained for long training times, 2) predict performance over long training times by using larger networks trained for just 1 epoch. See Appendix C for experimental details and Figures 10 and 11 for full results.

Figure 3 illustrates that scale-time equivalence can indeed be used to extrapolate performance on large scales and training times. Predictions of test and train performance under large model scales are particularly close to the true performance; notably, we can closely predict the scale at which generalization starts to occur. However, there is a small discrepancy between predicted and actual

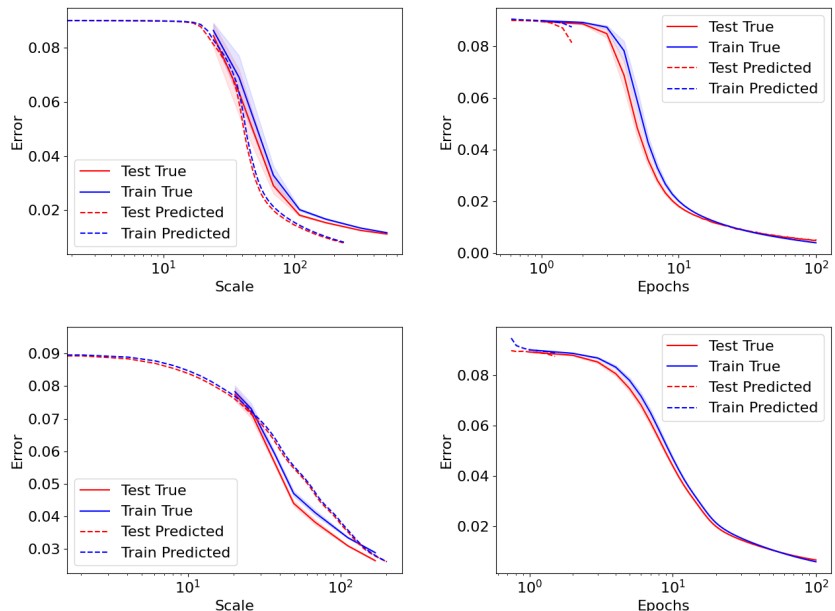

Figure 3: **By scale-time equivalence, small models trained for long times predict performance of large models trained for small times and vice versa: test of prediction.** Predicted and true test and train error of a CNN (top row) and MLP (bottom row) trained on MNIST. Column 1: predicting the performance of larger models over a few epochs by training smaller models for up to 100 epochs. Column 2: predicting performance of smaller models over many epochs by training larger models for 1 epoch. We use scale-time equivalence to predict the equivalent scale or number of epochs for each prediction. Margins indicate standard errors over 5 trials.

performance; we believe this can be corrected with dataset and model-specific tuning of the scale-time trade-off curve. Nevertheless, our findings reveal that scale-time equivalence can be used to predict optimal network scale and training time.

## 5  A UNIFIED VIEW OF DOUBLE DESCENT W.R.T. TRAINING TIME, MODEL SCALE AND TRAINING SET SIZE

We next leverage scale-time equivalence to obtain a more-unified understanding of the phenomenon of double descent, with respect to training time, parameter count and training set size.

### 5.1  ERROR SCALING OVER TIME IN A LINEAR MODEL

Following the approach of Pezeshki et al. (2022); Heckel & Yilmaz (2021); Stephenson & Lee (2021); Schaeffer et al. (2023), we first present a simple linear model that explains double descent with respect to time.

Consider a linear student-teacher setting in which training set outputs $Y \in \mathbb{R}^n$ are constructed as:

$$Y = Xw + \varepsilon \tag{5}$$

where training data $X \in \mathbb{R}^{n \times m}$, noise $\varepsilon \in \mathbb{R}^n$, $w \in \mathbb{R}^m$ is the unknown true model, $n$ is the number of training points and $m$ is the model dimensionality. We assume $w$ is drawn independently from $\varepsilon$. Then, the parameters $\theta_t$ of a linear model learned after $t$ time of gradient flow on mean squared error (with learning rate $\eta$) can be expressed as:

$$\theta_t = X^\dagger (I - e^{-\eta XX^T t}) Y \tag{6}$$

The resulting prediction error $x^T \theta_t - x^T w$ on a test point $x$ can be expressed as:

$$x^T [X^\dagger (I - e^{-\eta XX^T t}) X - I] w + x^T X^\dagger (I - e^{-\eta XX^T t}) \varepsilon \tag{7}$$

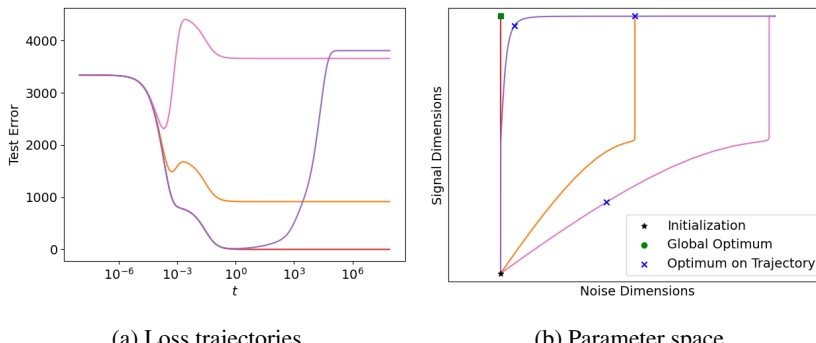

(a) Loss trajectories          (b) Parameter space

Figure 4: **Predicted variations in double-descent behavior, depending on training noise profile.** Schematic loss trajectories (a) and corresponding parameter space trajectories (b) of linear regression under various noise settings (different color curves). Depending on the noise profile, parameters may experience a temporary increase in error resembling an interpolation threshold.

We then use the singular value decomposition $U\Sigma V^T$ of $X$ to express the prediction error:

$$x^T V[\Sigma^\dagger (I - e^{-\eta \Sigma \Sigma^T t})\Sigma - I]V^T w + x^T V \Sigma^\dagger (I - e^{-\eta \Sigma \Sigma^T t})V^T \varepsilon$$
$$= \sum_{i=1}^{m} -(x^T V)_i (V^T w)_i e^{-\eta \sigma_i^2 t} + (x^T V)_i (V^T \varepsilon)_i \frac{1 - e^{-\eta \sigma_i^2 t}}{\sigma_i} \quad (8)$$

where $\sigma_i$ are the singular values of $X$, and we denote $\sigma_i = 0$ for $i > n$ when $n < m$ (in this case, $\frac{1-e^{-\eta \sigma_i^2 t}}{\sigma_i}$ denotes 0). Using the independence of $w$ and $\varepsilon$, we finally may simply express the expected *squared* prediction error as:

$$\mathbb{E}[(x^T w - x^T \theta_t)^2] = \mathbb{E}[(\sum_{i=1}^{m} S_i e^{-\eta \sigma_i^2 t})^2] + \mathbb{E}[(\sum_{i=1}^{m} N_i \frac{1 - e^{-\eta \sigma_i^2 t}}{\sigma_i})^2] \quad (9)$$

where $S_i = -(x^T V)_i (V^T w)_i$, $N_i = (x^T V)_i (V^T \varepsilon)_i$. The first, signal term captures how well $w$ can be learned in the absence of noise. In the underparameterized regime ($n > m$), this term approaches 0 as $t \to \infty$: without noise, the model can be learned perfectly. Observe that in general, the prediction error is *not* predicted to decay directly as a power law with $t$: instead, it decays or grows following a combination of exponential curves (though a combination of exponential decays at different rates can mimic a power law (Reed & Hughes, 2002)).

The second, noise term initially starts at 0 and grows over time. Notably, the size of the noise term is largest near the interpolation threshold (when $m \approx n$) since the smallest singular values $\sigma_i$ will take small, non-zero values. As $n$ or $m$ grows larger than the other (distance from the interpolation threshold increases), the size of the noise term decreases. However, the noise term's magnitude monotonically increases with $t$.

For any given singular component of $X$, if the noise components are large relative to signal components (i.e. $|N_i| > |S_i|$), we expect that error will increase at the corresponding timescale, around $t = \frac{1}{\eta \sigma_i^2}$ and vice versa. Thus, if the small singular value components are noisy, error will increase later during training, while if large singular value components are noisy, error will increase early during training. Double-descent occurs when noisy components are acquired first (increase in error) followed by signal components (decrease in error). Figure 4(a) illustrates how different acquisition rates of noise vs. signal components may yield different loss trajectories, with some corresponding to double descent. We may also view these trends geometrically in the parameter space; Figure 4(b) shows that if signal and noise correspond to orthogonal parameter dimensions, then double descent (orange curve) corresponds to a setting in which noise dimensions are learned rapidly before signal dimensions. The optimal training time depends on which point in the parameter trajectory is closest to the true optimum and may occur either at the end of training or at an intermediate point.

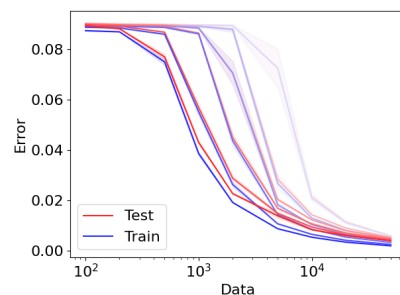

Figure 5: **Larger models require less data to interpolate: test of prediction.** Test and train mean squared error of CNN models trained on MNIST under varying levels of data. Different curves indicate different model scales; darker colors indicate larger models. Margins indicate standard errors over 5 trials.

## 5.2 EARLY NOISE ACQUISITION CAN EXPLAIN PARAMETER-WISE DOUBLE DESCENT

Next, we hypothesize that due to scale-time equivalence, parameter-wise double descent occurs *due to the same mechanism* as time-wise double descent. Namely, in settings where noise is acquired before signal, smaller scale models (which require effectively larger training time by scale-time equivalence) fail to acquire signal.

More precisely, by combining Equation 9 and scale-time equivalence, we propose the following scaling law in terms of the number of parameters $p$ and time $t$ by substituting $t$ with $pt$:

$$error^2 = \mathbb{E}[(\sum_{i=1}^{\infty} S_i e^{-\eta \sigma_i^2 pt})^2] + \mathbb{E}[(\sum_{i=1}^{\infty} N_i \frac{1 - e^{-\eta \sigma_i^2 pt}}{\sigma_i})^2] \tag{10}$$

where $\sigma_i$ depends on the number of training points $n$ and $S_i$ and $N_i$ are random variables determining the strength of signal and noise respectively. This scaling law simultaneously explains double descent in terms of both $p$ and $t$. Moreover, it can explain double descent in terms of data volume $n$ as well: if $n$ is set such that there are several small non-zero values of $\sigma_i$, the noise term becomes amplified. This explanation for double descent in $n$ follows prior literature (Advani & Saxe, 2017).

The explanation for parameter-wise double descent differs from conventional wisdom in which double descent occurs due to the same reason as double descent in $n$: namely, when the number of model parameters is close to $n$, the model is highly sensitive to noisy directions in the training data (corresponding to small $\sigma_i$ in a linear model), thus severely overfitting. How can we distinguish this hypothesis from ours? We propose three tests to separate the two hypotheses.

## 5.3 LESS DATA REQUIRED FOR GENERALIZATION WITH MODEL SCALE

The interpolation threshold can be defined as the point when the number of data points equals the effective complexity of a model (Nakkiran et al., 2019). Thus, for any model, the location of the interpolation threshold with respect to data volume is the model's effective capacity: higher capacity models have a rightward-shifted interpolation threshold with respect to data volume. Conventional double descent theory argues that this point corresponds to a sharp decrease in the test set error as the amount of data grows. Starting from zero data, we would therefore expect that larger models (which have a larger effective capacity), would experience a sharp decrease in test set error at *higher data volumes*: larger models require *more* data to generalize. By contrast, under our explanation, as model size grows, the amount of data needed to generalize *decreases*: since model size corresponds to training time, generalization occurs more easily with larger models (equivalently, more training time).

To test this, we conduct experiments on benchmark datasets and architectures. Figure 5 reveals that larger models indeed require less data to generalize, thus supporting our hypothesis. (See Appendix C for experimental details and Figure 8 for full results.) Indeed, the *training set errors decrease* with data volume which is at odds with conventional double descent theory in which more

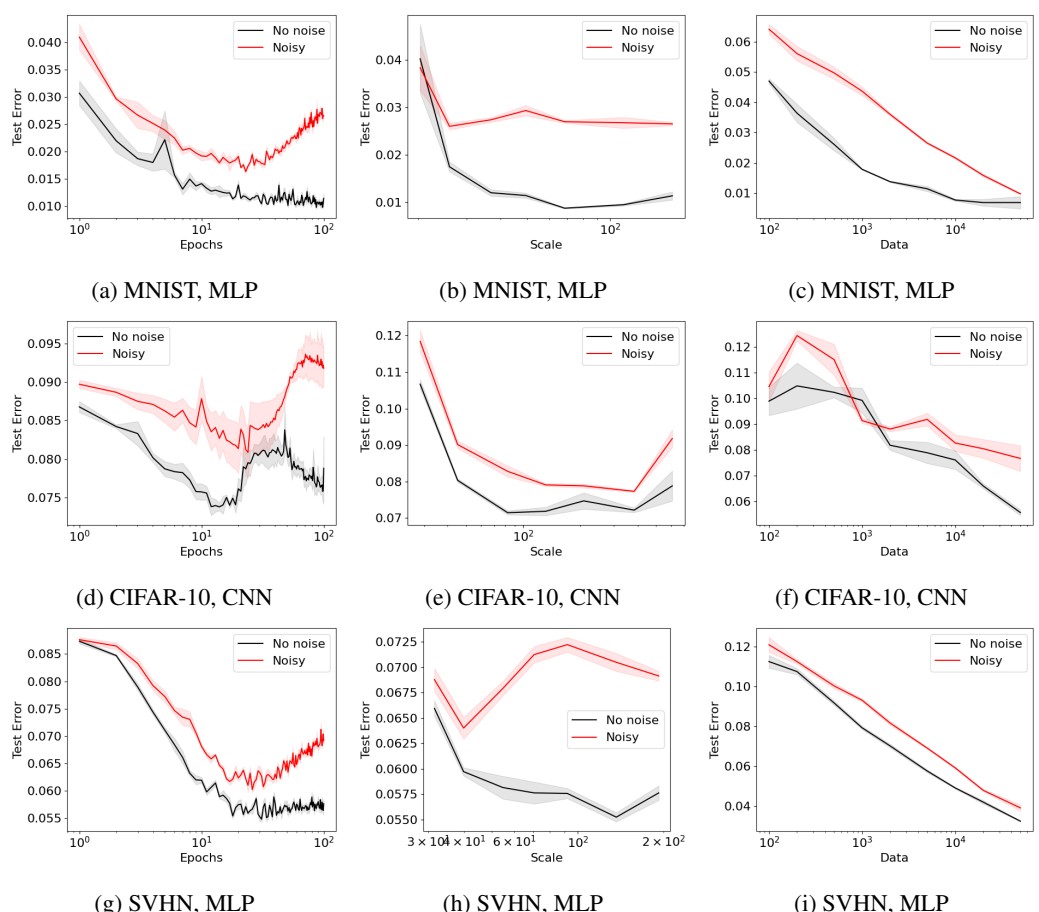

Figure 6: **Noise induces a persistent or growing error with training time and model scale, but not with dataset size.** Test mean squared error vs. number of epochs, model scale and training data under noisy and noise-free labels. Each row indicates a different combination of dataset and architecture. Margins indicate standard errors over 5 trials.

data is always harder to interpolate. Fundamentally, this is because conventional double descent theory typically assumes complete training convergence, which does not explain phenomena under the practically relevant setting of a fixed training budget. In summary, the increased ease of generalization with model scale does not neatly fit into standard theories of double descent, but readily fits in our explanation.

### 5.4 PERSISTENT VS LOCAL EFFECTS OF NOISE ON ERROR CURVES

Another key distinction is how the two explanations behave under varying levels of noise. Under the conventional explanation, as the noise ($|N_i|$) increases, the error increases mostly *locally around* $p \approx n$, near the interpolation threshold. This is because in Equation 9, the noise coefficients $N_i$ multiply terms that are largest near the interpolation threshold and decrease as either data volume or model scale grow larger than the other (Schaeffer et al., 2023). In contrast, under our hypothesis, as the noise increases, the error monotonically grows with both model scale and time (see Equation 10). In other words, when the noise level changes, our hypothesis predicts a global performance change with respect to model scale while the conventional explanation predicts a primarily local change.

We conduct experiments on benchmark datasets and architectures to validate this. We add label noise to the training set and evaluate performance as a function of epochs, model scale and data size. Appendix C includes experimental details. Noise leads to an increase in error in all cases, Figure 6 (See Appendix D for additional plots), with a persistent or growing error with training time (epochs)

and model scale. However, it leads to only localized increases in error in terms of the dataset size: in the case of MNIST and SVHN, for instance, there is nearly no change in error at the highest data volume.

### 5.5 SHAPE OF LOSS CURVE: UPTURN WITH MODEL SIZE

A final key distinction between the two explanations is their prediction of the shape of the loss curve. Under the conventional explanation of double descent, past the interpolation threshold, increasing model size always improves performance. However, under our hypothesis, increasing model size may in certain cases *worsen performance*, if signal features are learned before noisy features. In Figure 6, we find that CIFAR-10 and SVHN indeed exhibit a U-shaped error curve with respect to model scale such that larger models do worse. This departure from a conventional double descent curve is consistent with our model.

## 6 DISCUSSION

In this work, we demonstrate that model scale and training time can be traded off with each other. This enables us to re-frame parameter-wise double descent as occurring due to the same mechanism as epoch-wise double descent, which occurs due to the early acquisition of noise features during training. This framing of parameter-wise double descent has a number of surprising implications unexplained by standard explanations of double descent: 1) generalization requires less data with a larger model, 2) label noise significantly increases test error even for highly overparameterized models, 3) increasing model scale for overparameterized models need not always improve performance.

How can we reconcile our findings with conventional theory and experimental findings on double descent and scaling laws? Note that in past work, the presence of double descent with respect to model scale is often dependent on the choice of dataset, model, and whether label noise is added (Nakkiran et al., 2021). While past explanations of parameter-wise double descent necessitate that it must occur, our explanation is more flexible: double descent need not occur if noisy features are acquired later in training than signal features. Indeed, our theory explains why local increases in test error in terms of model scale are often relatively modest in contrast to the sharp spike predicted by conventional theory (Belkin et al., 2019). Thus, our explanation is more consistent with the variability of double descent observed in the literature.

Regarding scaling laws, our predicted scaling law in Equation 10 is more flexible than the power law scalings predicted in prior literature; indeed, with the proper settings of signal and noise parameters $S_i$ and $N_i$, we may recover power-law scalings of error with respect to time, model scale and data volume. However, our approach retains the flexibility to explain error scalings in settings where a power law *does not* explain empirical error trends, as we see in our experimental results.

Given the parametric flexibility of our scaling law, how can we use it to predict performance as model scale or training time increases? We demonstrate that by scale-time equivalence, performance under varying training times can be used to predict performance under varying model scales and vice versa. Our approach eliminates the need for strong parametric assumptions in the form of scaling law to make extrapolation predictions. This is particularly useful in cases where error *increases* with scale for overparameterized models since our approach can be used to predict an optimal model size. On the other hand, we require empirically evaluating the performance of models (under either a smaller scale or lower training time). We believe our approach can be valuable to practitioners who have the flexibility to run some limited empirical small-scale experiments before full-scale training. Finally, our results suggest that smaller models trained for a longer time may behave as well as larger models, which we empirically observe on vision benchmarks. This is particularly important in the age of LLMs, where very large models are trained for a small number of epochs (often just one).

We also highlight some important limitations of our work. Our experiments are all conducted on standard vision benchmarks; we believe testing our theory on the language domain is a critical future direction. Another limitation is that we do not yet have a fundamental understanding of what sets the appropriate model scale of a neural network. Experimentally, we found that the cube root of the number of model parameters is appropriate, but without a strong theoretical basis; this deserves further study. Overall, we believe our contributions not only shed light on neural scaling laws, but also present exciting directions for future work.

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

## A  PROOF OF THEOREM 1

First, observe that gradient descent applied to $\theta$ corresponds to:

$$\dot{\theta} = -\eta R^T K^T \nabla L(\alpha) \tag{11}$$

This yields a change in $\alpha$ of:

$$\dot{\alpha} = -\eta K R R^T K^T \nabla L(\alpha) \tag{12}$$

**Properties of** $KRR^T K^T$   First, we investigate the properties of the $r \times r$ random matrix $KRR^T K^T$. Since the elements of $R$ are drawn iid from a unit Gaussian, $\mathbb{E}[RR^T] = pI$. Thus,

$$\mathbb{E}[KRR^T K^T] = pKK^T \tag{13}$$

Next, consider the element at the $i$th row and $j$th column of $KRR^T K^T$. This may be expressed as:

$$K_{i,:}RR^T K_{:,j}^T = \sum_l K_{i,:}R_{:,l}R_{l,:}^T K_{:,j}^T = \sum_l (K_{i,:}R_{:,l})(K_{j,:}R_{:,l}) \tag{14}$$

where we express different columns of $R$ as $R_{:,l}$. Note that each term in the summand is independent from one another since $R$ has iid elements. Observe that $K_{i,:}R_{:,l} = \sum_k K_{i,k}R_{k,l}$ and each term $K_{i,k}R_{k,l}$ is an independent mean zero Gaussian with variance $K_{i,k}^2$. Thus, $K_{i,:}R_{:,l}$ is a mean zero Gaussian with variance $K_{i,:}K_{:,i}^T$. Moreover, since the expectation of $K_{i,:}RR^T K_{:,j}^T$ is $pK_{i,:}K_{:,j}^T$, $K_{i,:}R_{:,l}$ and $K_{j,:}R_{:,l}$ are jointly Gaussian with covariance $K_{i,:}K_{:,j}^T$.

Now, consider the expectation of $(K_{i,:}RR^T K_{:,j}^T)^2$:

$$\mathbb{E}[(K_{i,:}RR^T K_{:,j}^T)^2] = \mathbb{E}[\sum_l (K_{i,:}R_{:,l})(K_{j,:}R_{:,l}) \sum_{l'} (K_{i,:}R_{:,l'})(K_{j,:}R_{:,l'})] \tag{15}$$

since $(K_{i,:}R_{:,l})(K_{j,:}R_{:,l})$ and $(K_{i,:}R_{:,l'})(K_{j,:}R_{:,l'})$ are independent for $l \neq l'$:

$$\mathbb{E}[(K_{i,:}RR^T K_{:,j}^T)^2] = \mathbb{E}[\sum_l (K_{i,:}R_{:,l})^2(K_{j,:}R_{:,l})^2] + p(p-1)\mathbb{E}[(K_{i,:}R_{:,l})(K_{j,:}R_{:,l})]^2 \tag{16}$$

$\mathbb{E}[(K_{i,:}R_{:,l})(K_{j,:}R_{:,l})]$ is simply $K_{i,:}K_{:,j}^T$. Since $K_{i,:}R_{:,l}$ and $K_{j,:}R_{:,l}$ are jointly Gaussian, we may reparameterize them as:

$$K_{i,:}R_{:,l} = az_i \tag{17}$$

$$K_{j,:}R_{:,l} = bz_i + cz_j \tag{18}$$

where $z_i$ and $z_j$ are independent unit Gaussians, $a = \sqrt{K_{i,:}K_{:,i}^T}$, $b = \frac{K_{i,:}K_{:,j}^T}{a}$, $c = \sqrt{K_{j,:}K_{:,j}^T - b^2}$. Then, $(K_{i,:}R_{:,l})^2(K_{j,:}R_{:,l})^2$ may be expressed as:

$$(K_{i,:}R_{:,l})^2(K_{j,:}R_{:,l})^2 = (az_i)^2(bz_i+cz_j)^2 = a^2 z_i^2(bz_i+cz_j)^2 = a^2b^2 z_i^4 + 2a^2bc z_i^3 z_j + a^2c^2 z_i^2 z_j^2 \tag{19}$$

Taking the expectation:

$$\mathbb{E}\left[a^2b^2 z_i^4 + 2a^2bc z_i^3 z_j + a^2c^2 z_i^2 z_j^2\right] = a^2b^2\mathbb{E}[z_i^4] + 2a^2bc\mathbb{E}[z_i^3 z_j] + a^2c^2\mathbb{E}[z_i^2 z_j^2] \tag{20}$$

Using the moments of unit Gaussians, we have $\mathbb{E}[z_i^4] = 3$, $\mathbb{E}[z_i^3 z_j] = 0$, $\mathbb{E}[z_i^2 z_j^2] = 1$. Thus, we have

$$\mathbb{E}[(K_{i,:}R_{:,l})^2(K_{j,:}R_{:,l})^2] = 3a^2b^2 + a^2c^2 = a^2(3b^2 + c^2)$$

$$= a^2(K_{j,:}K_{:,j}^T + 2\frac{(K_{i,:}K_{:,j}^T)^2}{a^2}) = K_{i,:}K_{:,i}^T K_{j,:}K_{:,j}^T + 2(K_{i,:}K_{:,j}^T)^2 \tag{21}$$

Finally, we may express the expectation of $(K_{i,:}RR^T K_{:,j}^T)^2$ as:

$$\mathbb{E}[(K_{i,:}RR^T K_{:,j}^T)^2] = p(K_{i,:}K_{:,i}^T K_{j,:}K_{:,j}^T + 2(K_{i,:}K_{:,j}^T)^2) + p(p-1)(K_{i,:}K_{:,j}^T)^2 \tag{22}$$

**Decomposition into expectation and noise**  We now decompose $KRR^T K^T$ into an expectation term and a mean-zero noise term:

$$KRR^T K^T = pKK^T + N \tag{23}$$

where $N \in \mathbb{R}^{r \times r}$ is a mean-zero matrix. Note that each element of $N$ has variance:

$$\mathbb{E}[N_{i,j}^2] = \mathbb{E}[(K_{i,:}RR^T K_{:,j}^T)^2] - p^2(K_{i,:}K_{:,j}^T)^2 = p(K_{i,:}K_{:,i}^T K_{j,:}K_{:,j}^T + (K_{i,:}K_{:,j}^T)^2) \tag{24}$$

Thus, the squared Frobenius norm of $N$ has expectation:

$$\mathbb{E}[||N||_F^2] = \sum_{i,j} \mathbb{E}[N_{i,j}^2] = p\sum_{i,j} K_{i,:}K_{:,i}^T K_{j,:}K_{:,j}^T + (K_{i,:}K_{:,j}^T)^2 = p(||K||_F^4 + ||KK^T||_F^2) \tag{25}$$

We express the dynamics of $\alpha$ as:

$$\dot{\alpha} = -\eta p KK^T \nabla L(\alpha) - \eta N \nabla L(\alpha) \tag{26}$$

Now suppose we have two copies of $\alpha$: one copy ($\alpha^{(1)}$) with noise-free dynamics, and a second copy with noise:

$$\dot{\alpha}^{(1)} = -\eta p KK^T \nabla L(\alpha^{(1)}) \tag{27}$$

$$\dot{\alpha}^{(2)} = -\eta p KK^T \nabla L(\alpha^{(2)}) - \eta N \nabla L(\alpha^{(2)}) \tag{28}$$

We define the discrepancy between them as $\delta = \alpha^{(2)} - \alpha^{(1)}$, which has dynamics:

$$\dot{\delta} = -\eta p KK^T [\nabla L(\alpha^{(2)}) - \nabla L(\alpha^{(1)})] - \eta N \nabla L(\alpha^{(2)}) \tag{29}$$

Consider the rate of change of the $\ell_2$ norm of $\delta$:

$$\frac{d}{dt}||\delta|| \leq ||\dot{\delta}|| \leq \eta p ||KK^T[\nabla L(\alpha^{(2)}) - \nabla L(\alpha^{(1)})]|| + \eta||N \nabla L(\alpha^{(2)})|| \tag{30}$$

Using the Lipschitz bounds on $\nabla L$ and $L$:

$$\frac{d}{dt}||\delta|| \leq \eta p h ||KK^T|| ||\delta|| + \eta ||N|| l \tag{31}$$

where the matrix norms in the expression denote $\ell_2$ operator norm.

Now, returning to $||N||_F^2$, note that by Markov's inequality, with probability $1 - \epsilon$:

$$||N||_F^2 \leq \frac{p}{\epsilon}(||K||_F^4 + ||KK^T||_F^2) \tag{32}$$

Using the fact that $||N|| \leq ||N||_F$, with probability $1 - \epsilon$:

$$\frac{d}{dt}||\delta|| \leq \eta p h ||KK^T|| ||\delta|| + \eta \frac{\sqrt{p}}{\sqrt{\epsilon}}\sqrt{||K||_F^4 + ||KK^T||_F^2} l \tag{33}$$

This is a differential inequality in $||\delta||$. Observe that $||\delta||$ takes its maximum possible trajectory at equality. Assuming $||\delta|| = 0$ at time $t = 0$, this differential inequality may be solved by simply solving the equality case and setting the solution as the upper bound on $\delta_t$ (with subscript denoting time $t$):

$$||\delta_t|| \leq \frac{\eta \frac{\sqrt{p}}{\sqrt{\epsilon}}\sqrt{||K||_F^4 + ||KK^T||_F^2} l}{\eta p h ||KK^T||}(e^{\eta p h ||KK^T|| t} - 1) = \frac{\sqrt{||K||_F^4 + ||KK^T||_F^2} l}{\sqrt{p\epsilon} h ||KK^T||}(e^{\eta p t h ||KK^T||} - 1) \tag{34}$$

**Solving the noise-free dynamics**  Now, we consider the noise-free dynamics:

$$\dot{\alpha}^{(1)} = -\eta p KK^T \nabla L(\alpha^{(1)}) \tag{35}$$

Note that $A_{pt}$ solves the dynamical equation for $\alpha^{(1)}$ by the chain rule:

$$\frac{d}{dt}A_{pt} = p(-\eta KK^T \nabla L(A_{pt})) = -\eta p KK^T \nabla L(A_{pt}) \tag{36}$$

Thus, we may simply express $\alpha_t^{(1)}$ as $A_{pt}$.

**Final result**    Combining and reexpressing our previous results, we may write that with probability $1 - \epsilon$:

$$||\alpha_t^{(2)} - A_{pt}|| = ||\alpha_t^{(2)} - \alpha_t^{(1)}|| = ||\delta_t|| \leq \frac{l\sqrt{||K||_F^4 + ||KK^T||_F^2}}{h\sqrt{p\epsilon}||KK^T||}(e^{\eta pth||KK^T||} - 1) \quad (37)$$

Thus, the true dynamics deviate from $A_{pt}$ by a bounded amount.

## B    MEASURING EFFECTIVE PARAMETER COUNT

In this section, we aim to provide a justification for why the *cube root* of the absolute number of parameters of a network is a good proxy for the *effective* number of parameters. Finding the effective number of parameters requires defining the parameter count at which the interpolation threshold occurs; this number is the effective parameter count. Unfortunately, in most models, training data is not *perfectly* interpolated by the model; thus defining the interpolation threshold location exactly is nontrivial. Instead, we identify a property of the interpolation threshold in the training error of linear models, then identify at which parameter count this property also holds in nonlinear models.

**Training error in linear models**    Suppose we are provided a training set $X \in \mathbb{R}^{n \times p}$ and corresponding training labels $Y \in \mathbb{R}^n$ where $n$ is the number of data points. Assume that $X$ has elements drawn uniformly from a unit Gaussian. We aim to find a parameter $\theta \in \mathbb{R}^p$ such that:

$$Y \approx X\theta \quad (38)$$

The minimum norm solution minimizing the mean squared error is:

$$\theta = X^\dagger Y \quad (39)$$

where $\dagger$ denotes pseudoinverse. Denoting the predicted training labels as $\hat{Y}$, the mean squared error is then:

$$\frac{1}{n}||Y - \hat{Y}||_2^2 = \frac{1}{n}||Y - XX^\dagger Y||_2^2 = \frac{1}{n}Y^T(I - XX^\dagger)Y \quad (40)$$

Finally, assuming that $Y_i^2 = 1$ for all $i$, we may write the mean training error as:

$$\frac{1}{n}||Y - \hat{Y}||_2^2 = \max(1 - \frac{p}{n}, 0) \quad (41)$$

Thus, for a linear model, the training error decreases linearly at rate $-\frac{1}{n}$ with respect to $p$ before the interpolation threshold ($p = n$), and then is zero after the interpolation threshold. We extract a key property around the interpolation threshold from the linear model: the training error is $\frac{1}{n}$ at $p = n - 1 = O(n)$. Note that for any $\alpha < 1$, the training error is $\frac{1}{n^\alpha}$ *before* $O(n)$.

**Power law training rate decay**    Next, we consider power-law decays of training rate error and characterize which power laws are consistent with the interpolation threshold properties outlined above. Consider a power law decay of the mean training error of $\frac{n^\alpha}{p^\beta}$. Note that in order to satisfy condition (2), we must have:

$$\frac{1}{n} = O(\frac{n^\alpha}{n^\beta}) \quad (42)$$

where we set $p = O(n)$. Thus, $\beta - \alpha = 1$. Power-law decays of the form $\frac{n^\alpha}{p^{\alpha+1}}$ are consistent with the interpolation threshold property.

**Error scaling in neural networks**    Next, we turn to model training error scaling in neural networks. We make the following heuristic argument: to fit $n$ training points, a network of width $m$ needs to encode $O(mn)$ numbers corresponding to $m + 1$ numbers for each training point (to represent the features and labels of each training point). On the other hand, the network has $O(m^2)$ parameters since its intermediate layer weights have $m^2$ parameters. We expect that scaling both the network's capacity of $O(m^2)$ and the required capacity of $O(mn)$ at the same rate will not change the training error. Thus, we expect the training error to be a function of $\frac{mn}{m^2} = \frac{n}{m}$.

To find *which* function of $\frac{n}{m}$ models the training error, we introduce another argument. We assume that the network can be approximated as an ensemble of $O(m^2)$ submodels, each with $O(1)$ parameters. Suppose the model output is the average of the outputs of the submodels. Then, assuming that the distribution of the submodel outputs on any given training point has variance $O(1)$ over different model initializations, the variance of the ensemble output is $O(\frac{1}{m^2})$. Thus, over model initialization, the model output at any given training point will center around a mean value with deviations on the order of $O(\frac{1}{m})$. Assuming that as the network capacity goes to $\infty$, the training error goes to 0, the mean value of the predicted output on a training point must be the true value; thus, the predicted output differs from the true output by $O(\frac{1}{m})$. This corresponds to a mean squared error on the training points scaling as $O(\frac{1}{m^2})$. Finally, to get the dependence on $n$, we use the observation that the error must depend on only the fraction $\frac{n}{m}$. Thus, the mean squared training error scales as $O(\frac{n^2}{m^2})$.

Finally, we know by the argument above that power law error rate scaling of the form $\frac{n^\alpha}{p^{\alpha+1}}$ are consistent with the interpolation threshold property, where $p$ is the effective number of parameters. Equating this with the neural network scaling result of $O(\frac{n^2}{m^3})$, we must have that $\alpha = 2$, yielding:

$$O(\frac{n^2}{p^3}) = O(\frac{n^2}{m^2}) \tag{43}$$

Equating the denominators, we have $p^3 = m^2$, or $p = m^{2/3}$. In other words, the effective number of parameters is $m^{2/3}$. Since the absolute number of parameters in the network is $m^2$, the effective number of parameters is the cube root of the absolute parameter count.

## C  EXPERIMENTAL DETAILS

### C.1  MODEL TRAINING

We used three benchmark datasets: CIFAR-10, MNIST, and SVHN. Each dataset was subject to preprocessing involving standard normalization. For CIFAR-10 and SVHN, the normalization was performed using means and standard deviations of (0.5, 0.5, 0.5). For MNIST, the normalization used a mean and standard deviation of 0.5.

Two types of neural network architectures were evaluated:

- Convolutional Neural Network (CNN): Our CNN architecture consisted of six ReLU activated convolutional layers, with kernel sizes (3, 3, 3, 3, 3, 2 for CIFAR-10/SVHN or 1 for MNIST), strides (2, 1, 2, 1, 1, 1), and number of filters (5s, 10s, 20s, 40s, 80s) where s is a width parameter. This was followed by a fully connected layer.

- Multilayer Perceptron (MLP): The MLP architecture comprised six fully connected layers with hidden layer width 10s where s is a scale parameter.

The width parameter was set to values of 1, 2, 5, 10, 20, 50, and 100. When we refer to "model scale", we quantify this as the cube root of the number of network parameters rather than the value of the width parameter. We tested two learning rates (0.001 and 0.01) in combination with two optimizers (Adam and SGD respectively). Label noise was introduced at levels of 0.0 (no noise) and 0.2 (20% noise) to evaluate the robustness of the models. The number of training samples was varied among 100, 200, 500, 1000, 2000, 5000, 10000, 20000, and 50000. The samples were randomly selected from their respective base training sets. A constant batch size of 32 and 100 training epochs were used across all experiments. Mean Squared Error (MSE) was used as the loss function. Model performance was assessed using Mean Squared Error (MSE) on both training and test sets. For reproducibility, we set a manual seed for the PyTorch random number generator. Five different seeds (101, 102, 103, 104, 105) were used to assess the variance in results due to random initialization.

### C.2  SCALE-TIME TRADEOFF VALIDATION

**Linear model**  We construct a loss function as:

$$L(\theta) = ||K(R\theta + \beta_0) - \alpha^*||^2 \tag{44}$$

where $\alpha^*$ is a target value of $\alpha$. $K$, $R$, $\beta_0$ and $\alpha^*$ are all independently sampled from unit Gaussians, and $\theta$ is initialized from a unit Gaussian. We set $P = 1000$ and $r = 3$. We train $\theta$ via gradient descent with learning rate $10^{-6}$ and evaluate the number of iterations required to reach various values of the loss for varying values of $p$. This setup corresponds to a linear classifier trained with gradient descent on mean squared error using a training batch of $k = 3$ points where only $p$ of the $P = 1000$ parameters are controllable.

**Neural networks**  To evaluate the scale time tradeoff, for each model, we compute the minimum number of epochs necessary to achieve a test set MSE below 0.09.

### C.3  SHIFTING INTERPOLATION THRESHOLD

Experimental results are shown for networks trained with SGD.

### C.4  IMPACT OF NOISE AND SHAPE OF LOSS CURVE

Experimental results are shown for networks trained with Adam; since Adam trains faster, this allows us to examine training trends at later points in training.

### C.5 OPTIMAL SCALE PREDICTIONS

To predict error over scale using error over time or vice versa, we treat the quantity $pt$ as a predictor of performance, where $p$ is the effective number of network parameters. Thus, if we wish to know the error of a network at $(p_1, t_1)$ and we know the error of the network for all $p_0$, then we may predict the error at $(p_1, t_1)$ as the error at $(p_0, (\frac{p_1}{p_0})t_1)$.

### C.6 COMPUTING INFRASTRUCTURE

Experiments were run on a computing cluster with GPUs ranging in memory size from 11 GB to 80 GB.

# D    ADDITIONAL EXPERIMENTS

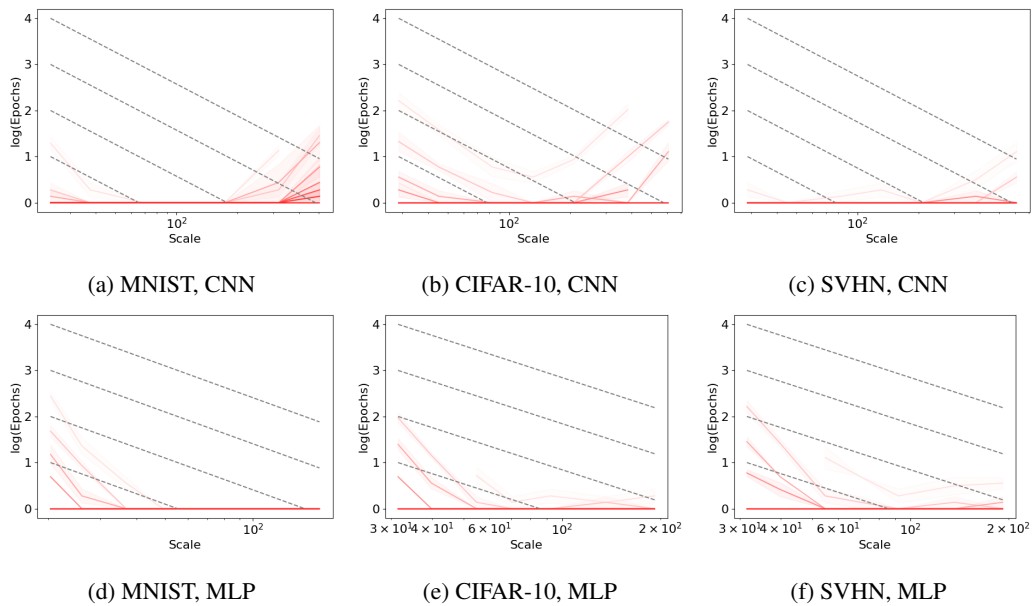

| | | |
|---|---|---|
| (a) MNIST, CNN | (b) CIFAR-10, CNN | (c) SVHN, CNN |
| (d) MNIST, MLP | (e) CIFAR-10, MLP | (f) SVHN, MLP |

Figure 7: Red lines indicate tradeoff curves between number of training epochs and network scale for different datasets and architectures trained with Adam. Different curves indicate different amounts of training data darker lines indicate more data. Curves are computed by, for each network scale, measuring the minimum amount of training time necessary to achieve non-zero generalization. Margins indicate standard errors over 5 trials. Grey curves are lines of 1:1 proportionality between scale and training epochs.

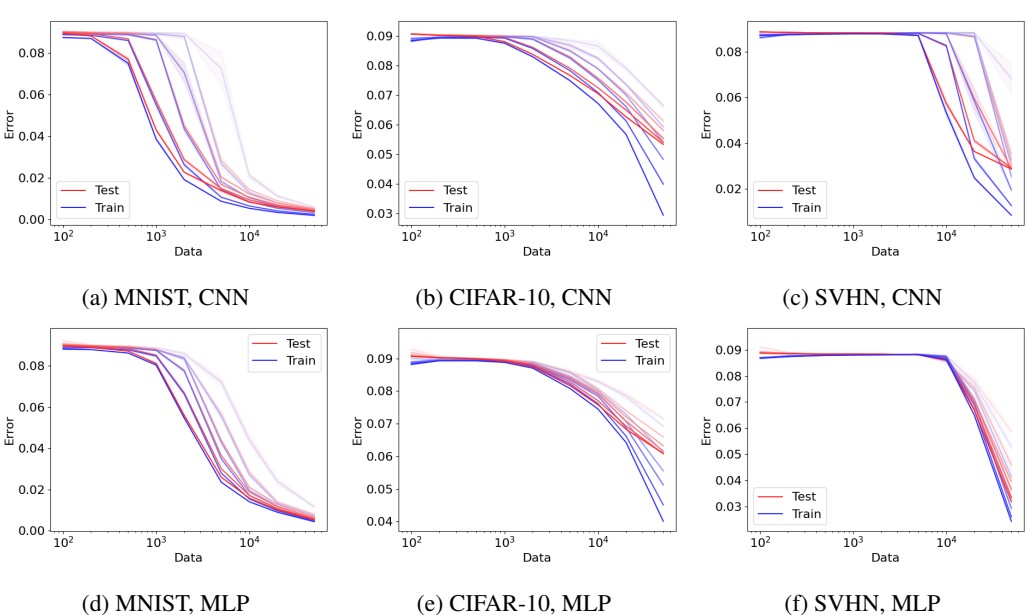

Figure 8: Test and train mean squared error of MLP and CNN models trained on benchmark datasets under varying levels of data. Different curves indicate different model scales; darker colors indicate larger models. Margins indicate standard errors over 5 trials.

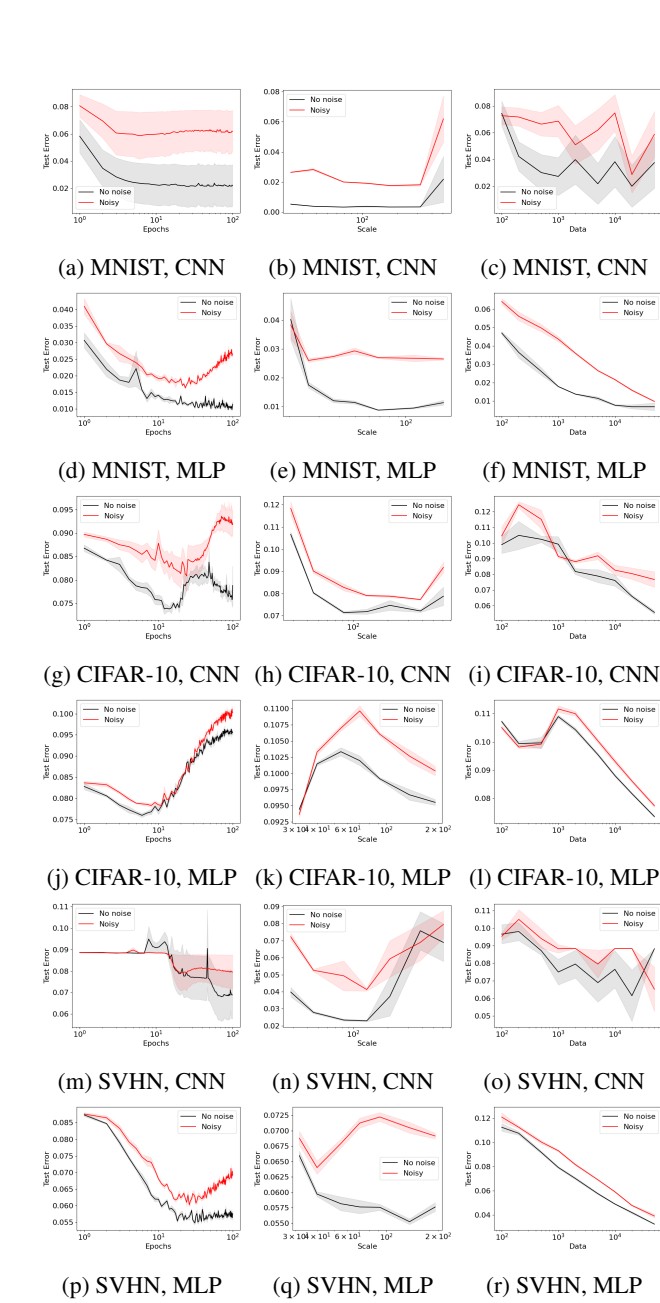

Figure 9: Test mean squared error vs. number of epochs, model scale and training data under noisy and noise-free labels. Each row indicates a different combination of dataset and architecture. Margins indicate standard errors over 5 trials.

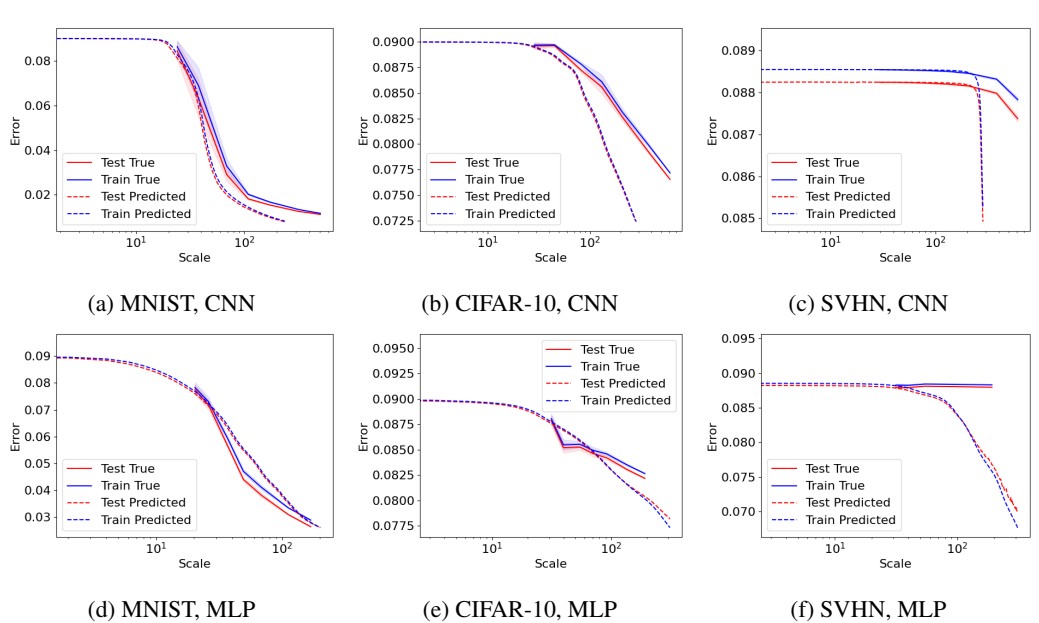

Figure 10: Predicted and true test and train mean squared error of MLP and CNN models trained on benchmark datasets under varying model widths for 10 epochs. Margins indicate standard errors over 5 trials. Predictions are generated by training a small model for 100 epochs and using scale-time equivalence to predict the equivalent scale.

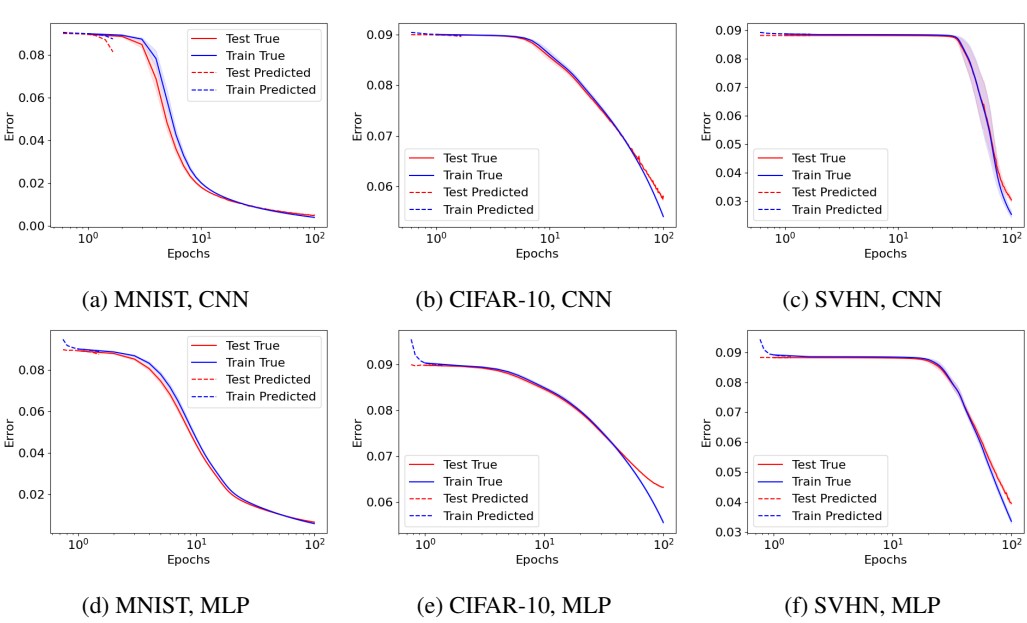

Figure 11: Predicted and true test and train mean squared error of MLP and CNN models trained on benchmark datasets under over training time for a medium-sized model. Margins indicate standard errors over 5 trials. Predictions are generated by training models of varying sizes for 1 epoch and using scale-time equivalence to predict the equivalent number of training epochs.

