# OpenReview forum: "Unified Neural Network Scaling Laws and Scale-time Equivalence"
_ICLR.cc/2025/Conference — Submitted to ICLR 2025_

### Official Review · Reviewer_W5rn · 2024-10-28

**Soundness:** 2
**Presentation:** 3
**Contribution:** 2
**Rating:** 3
**Confidence:** 3

**Summary:**

This work presents theoretical and empirical characterizations of how the model size, training time and data size affect the performance of neural networks. The authors first establish a scale-time equivalence in a linear model trained with gradient flow. They then conduct simulations on MNIST to verify their finding. Moreover,  the authors combine the scale-time equivalence with an analysis of the linear model to provide new interpretations of previously unexplained phenomena in double descent. Simulation results on CNN and MLP corroborate their interpretations.

**Strengths:**

The paper provides novel explanations to double descent and other unexplained phenomena based on analysis of linear models.

**Weaknesses:**

The theoretical derivations in this work are limited to simple linear models. It is unclear to what extent these observations hold in large models and other optimization settings.

* For example, Theorem 1 assumes the model is trained via gradient flow (which is close to full-batch GD), while in practice it is recommended to use one-pass SGD/Adam. Theorem 1 seems to lack a characterization of how the sample size correlates with the training time. This differs from the setting under which the authors conduct the simulations (mini-batch SGD).

* Also, the scaling laws (eq 9,10) are derived for linear models. However, the discussions in section 5.2-5.5 are for more general models, and are heuristically built upon these equations and some implicit assumptions on the signal and noise components. There lack precise and rigorous characterizations of the test error for nonlinear models/neural networks.

* Lack a discussion of more recent works on explaining neural scaling laws: e.g., [1,2]. In particular, [1] studied the scaling laws in linear model trained via one-pass sgd. I wonder how results in [1] reconcile with Theorem 1 in your work.

* I would appreciate an explicit discussion of the limitations of this work.





[1]. Lin, L., Wu, J., Kakade, S. M., Bartlett, P. L., & Lee, J. D. (2024). Scaling Laws in Linear Regression: Compute, Parameters, and Data. arXiv preprint arXiv:2406.08466.

[2]. Paquette, E., Paquette, C., Xiao, L., & Pennington, J. (2024). 4+ 3 Phases of Compute-Optimal Neural Scaling Laws. arXiv preprint arXiv:2405.15074.

**Questions:**

Apart from the points in weakness, here are my additional questions.

* In Figure 2, how is the batch size chosen? Would different choices of batch size affect the result?

* How is eq 10 derived from eq 9? My understanding is that M can be interpreted as the model size in linear models. Thus, eq 10 corresponds to the case $M\to\infty$ and it is unclear what the parameter size $p$ is.

---

> ### Author Response · Authors · 2024-11-22
>
> We thank the reviewer for their insightful comments and for identifying areas where our manuscript can be improved. We address the concerns below.
>
> **Generality of the Theoretical Analysis**
>
> Please refer to our general response, where we clarify the scope of our theoretical model and its relevance to neural networks. While our analysis uses gradient flow (akin to full-batch gradient descent), we believe that the insights gained are valuable for understanding fundamental aspects of neural network training dynamics.
>
> We acknowledge that extending the analysis to stochastic gradient descent (SGD) and larger models is important. Although our experiments use mini-batch SGD, prior work has shown that certain theoretical findings under gradient flow can provide useful approximations for SGD behavior, especially when the noise introduced by mini-batches averages out over time.
>
> **Scaling Laws and Applicability to Nonlinear Models**
>
> In section 5, we first present existing analyses of scaling laws with respect to training time in the context of linear models. These scaling laws have already been shown to be empirically effective in neural network models.
>
> Then, by scale-time equivalence, we apply these analyses to scaling with respect to model scale. Importantly, scale-time equivalence does not impose a linearity assumption; rather, the existing theoretical analysis uses a linear model.
>
> **Related Work**
>
> We appreciate the reviewer bringing to our attention recent works on neural scaling laws, such as Lin et al. (2023) and Paquette et al. (2023). We have added these papers in the revised manuscript.
>
> Unlike Lin et al. (2023), who study scaling laws in linear regression trained via one-pass SGD, our work focuses on the scale-time equivalence in the context of neural networks trained with gradient-based methods. We believe our findings complement this study by providing a theoretical analysis in more general, nonlinear models.
>
> **Limitations of the Work**
>
> We have added a dedicated section in the discussion to address the limitations of our work. We acknowledge that:
>
> - Our experiments are conducted on vision tasks with relatively small models and datasets. Extending the analysis to larger models and diverse tasks is an important direction for future research.
> - We currently lack a complete theoretical understanding of the cube root scaling observed in our experiments.
>
> By explicitly stating these limitations, we aim to provide a balanced perspective on our contributions.
>
> **Additional Questions**
>
> - Figure 2 Batch Size: In Figure 2, the batch size is fixed at 32, which is a standard choice in neural network training. We used a consistent batch size across experiments to isolate the effects of model scale and training time.
> - Derivation of Equation (10): Equation (10) is derived from Equation (9) simply by substituting $t$ with $pt$ to reflect the scale-time equivalence. We are unfortunately unsure what $M$ you are referring to but are happy to provide further clarification.

---

> > ### Comment · Reviewer_W5rn · 2024-11-23
> > **Clarification: derivation of Equation (10)**
> >
> > Thanks for the response! I was referring to $m$ in Eq. (9). How is $m$ related to $p$ in Eq. (10)?

---

> > > ### Author Response · Authors · 2024-11-23
> > >
> > > Thank you for your timely follow-up.
> > > In Equation (9) of our manuscript, the variable $m$ represents the model size, specifically the number of parameters in the linear model we consider. This equation provides a scaling law that describes how the test error depends on three key factors:
> > >
> > > - Training time $t$
> > > - Model dimensionality $m$
> > > - Number of data points $n$ (through the singular values $\sigma_i$​)
> > >
> > > This scaling law captures the relationship between these factors in linear models and has been shown to predict test error dynamics with respect to training time even in nonlinear models. However, we hypothesize that the existing scaling law is *not* a good model of scaling with respect to model size $m$ for nonlinear models. This is because it violates scale-time equivalence: $m$ and $t$ do not affect performance in the same way.
> > >
> > > In transitioning to Equation (10), our goal is to modify this equation by incorporating the concept of scale-time equivalence. To do this, we make two changes:
> > >
> > > - Setting $m \to \infty$: we believe the original $m$ in the equation does not correspond to model size in nonlinear models, and we set it to $\infty$
> > >
> > > - Applying Scale-Time Equivalence: based on our hypothesis of scale-time equivalence, we posit that increasing the model size has an effect equivalent to increasing the training time $t$ proportionally. Therefore, we replace the training time $t$ in Equation (9) with the product $pt$ in Equation (10), where $p$ represents model size. We use a different variable $p$ instead of $m$ to avoid confusion, although both refer to model size under different scaling laws.
> > >
> > > We hope this clarifies how $M$ and $p$ are related between the two equations and how we derived Equation (10) from Equation (9) using the scale-time equivalence. Please let us know if you have any further questions or need additional clarification.

---

### Official Review · Reviewer_wzg2 · 2024-11-02

**Soundness:** 2
**Presentation:** 2
**Contribution:** 2
**Rating:** 3
**Confidence:** 3

**Summary:**

The paper aims to build a theoretical model for the scaling laws of neural networks as a function of model size, training time and data volume. The focus of the paper is to show the existence of “scale-time equivalence”, meaning that in the training of a neural network, one can trade off parameters for training smaller models on more data. The authors also analyse double descent in the framing of their linear model, proposing that epoch-wise double descent happens when small models learn noisy features first from the data.

**Strengths:**

The paper studies an interesting phenomenon, relevant to practitioners and in line with the current trend in literature of scaling laws: can we train smaller models on more data without sacrificing performance? The authors also test their claims empirically on multiple datasets and architectures.

**Weaknesses:**

To begin with, in Section 3.1, the authors introduce a random subspace model as a toy setting to study their scaling argument. Could the authors explain why this model is a good approximation for what happens in the case of a neural network, as well as provide citations to prior works using a similar model (i.e. in line 127)? For example where does the input data get used in this model? What is the loss used to train model? It would help for the authors to expand with more details the setup they are working with.

My biggest critique however is that it seems to me that the bound that the authors get in Equation (4) is vacuous, for the following reason. Naively, the rhs of (4) seems to be dominated by $e^{pt}$ term. Thus, if we increase the number of parameters (or the training time, or both), the distance between $\alpha_t$ and $A_{pt}$ is upper bounded by a quantity that grows exponentially fast. Could the authors explain more what they mean in this case? Following the proof, I do not understand where $A_{pt}$ appears from in Eq. (36). Could the authors please elaborate?

It would be very helpful if the authors could provide more information with respect to the empirics. Several plots tend to be vague i.e. “darker lines indicate a smaller error threshold” - what is the actual value?, “darker lines indicate more data” - same question.

Continuing to Section 4, Figure 3 Column 2 seems to have incomplete traces for the “predicted” (dashed) lines. Could the authors explain what happened in this case?

In Section 5, the analysis of the linear model seems very similar to the analysis done in the seminal work of Advani and Saxe. While the authors rightfully cite this work, it is not clear to me what their analysis provides on top of the existing work.

**Questions:**

What are the empirical details used to run Figure 4?

Conceptually, I am unsure of the claim in line 505: “model scale and training time can be traded off with each other.” Suppose I take an overparameterized model with $p$ parameters that is able to interpolate the data in a fixed number of training steps $t$. Based on this claim, I should be able to make $p$ much smaller, up to the point of making the model underparameterized, if I increase $t$ proportionally - however, in this case, my model will not be able to train to convergence.

If the authors are open to discussion and to provide more details on their setup, I am willing to increase my score.

---

> ### Author Response · Authors · 2024-11-22
>
> We thank the reviewer for their constructive feedback and for pointing out areas where our manuscript could be enhanced. We address the concerns below.
>
> **Comments on the Random Subspace Model**
>
> Please refer to our general response, where we clarify the motivation and applicability of our theoretical model. Our random subspace model is designed to abstract key aspects of neural network training, such as overparameterization and the effects of increasing model size. By considering a general differentiable loss function and a parameterization that reflects the addition of parameters in random directions, we aim to capture the essence of neural network behavior.
>
> **Potential Vacuity of the Bound in Theorem 1**
>
> The reviewer raises a concern regarding the interpretation of the error bound in Equation (4). The bound contains an exponential term $e^{\eta p t h \| K K^\top \|}$, which suggests that the error could grow exponentially with $p$ and $t$. However, it is important to note that $pt$ represents the product of model scale and training time, which we interpret as the total training effort or progress.
> Our theorem shows that as $p$ increases for a fixed $pt$, the error bound decreases, indicating that the scale-time equivalence becomes more precise with larger $p$. The exponential term reflects the sensitivity of the training trajectory to initial conditions over longer training times, which is a common characteristic in dynamical systems, and not reflective of the degree of scale-time equivalence.
>
> **Clarification on Equation (36)**
>
> Equation (36) in the appendix follows from substituting $pt$ for $t$ in Equation (3)​. We are happy to provide further clarification if we have misunderstood the reviewer's question.
>
> **Experimental Details and Figure Clarity**
>
> Regarding the plots with graded colors, the specific choices of hyperparameters used to generate each line are detailed in Appendix C. We are happy to include these hyperparameters in the caption or change the format of the plot if another visualization would be more helpful.
>
> **Figure 3 Column 2**
>
> In Figure 3 Column 2 shows the predicted performance of models trained for many epochs based on training larger models for a limited number of epochs. The dashed lines represent the predictions, and they are truncated because, in practice, we are limited by computational resources and cannot train arbitrarily large models.
>
> **Comparison with Advani and Saxe**
>
> We appreciate the reviewer noting the similarity to the work of Advani and Saxe (2017). In fact, in Section 5, we take a similar analysis of scaling with respect to training time in linear models, and then, using scale-time equivalence, apply it to make novel predictions about scaling with respect to model size. We are happy to further elaborate on the relationship with this work.
>
> **Questions on Figure 4**
>
> Figure 4 is a schematic illustration intended to conceptually demonstrate how different noise profiles can affect loss trajectories and potentially lead to double descent behavior. It is not derived from empirical data but serves as a visual aid to support the discussion.
>
> **Concerns on Scale-Time Equivalence**
>
> The reviewer raises an interesting question about whether scale-time equivalence holds when the model size becomes very small, potentially making the model underparameterized. First, ur theoretical analysis indicates that the equivalence may break down for small model scale $p$, as reflected in the error bound of Theorem 1, which becomes looser when $p$ is small.
>
> However, assuming scale-time equivalence does hold, then if a large model is able to interpolate the data in finite time, so should a small model. On the other hand, it seems surprising that a small model would be able to fit the training data. How can we explain this seeming contradiction?
>
> In regimes where scale-time equivalence holds, the appropriate measure of capacity is the product $pt$, representing the total training effort, rather than model scale $p$. Therefore, a smaller model trained for a proportionally longer time has enough capacity and can achieve similar performance to a larger model trained for a shorter time, provided that $pt$ remains sufficient for learning the data.

---

> > ### Comment · Reviewer_wzg2 · 2024-11-24
> > **Reply**
> >
> > I'd like to thank the authors for their reply to my comments. I still have remaining questions that have not been answered:
> > - To begin, I did go through the global reply containing the explanation for the network structure and I still do not understand the setup. Could you please explain more in detail what this setup entails? For example where does the input data come into play? What is the relationship to a regular neural network? I am familiar with random feature models and I can sort of see a similarity between your setup and an RF model, but I am still quite confused and would need more explanations in order to move forward with my replies.
> >
> > - The bound still seems almost vacouous to me. Please correct me if I am wrong, your bound behaves as $\propto \exp(pt)/p^{0.5}$. If you keep $pt$ constant and increase $p$ then I do see that the bound should become tighter. But if you keep $pt$ constant and increase $t$, then your bound again becomes loose because $p$ would have to go down in order for the product to be constant. Should it not hold in both directions?
> >
> > - With respect to my last question from the original review, I am still currently confused. Namely, if I start with a model with $p$ params, trained for $t$ steps on $D$ data points (for sake of explanation, assume that the latent dimension of the data is also $D$), if $p > D$ this can interpolate the data. Now if I make $p$ much smaller (to the point of $p<D$) and I increase $t$ correspondingly, then I will have an irreducible loss. It seems to me that the scale-time equivalence fails in this case.
> >
> > Before I proceed to the rest of my comments, I would be grateful if the authors could reply to my current questions. Thank you very much!

---

> > > ### Author Response · Authors · 2024-11-25
> > >
> > > Thank you for your follow-up questions. We appreciate the opportunity to clarify our theoretical setup and address your concerns.
> > >
> > > **Clarification of the Theoretical Setup**
> > >
> > > Our theoretical model is designed to abstract key aspects of neural network training, focusing on the relationship between model size, training time, and learning dynamics. In this setup, we consider a parameter vector $\theta$ that represents the parameters of a neural network. The function implemented by the network depends on a projection of $\theta$ computed as $K R \theta + K \beta_0$.
> > >
> > > While we do not explicitly model the input data in our equations, the loss $L(\alpha)$ implicitly depends on the training data distribution. Our model abstracts away the specifics of the input data to focus on how the parameters $\theta$ and their effective representation $\alpha$ evolve during training.
> > >
> > > The assumption that we control a random $p$-dimensional subspace of a larger set of parameters $\beta$ reflects the practical scenario of increasing or decreasing a neural network's width or depth. Starting from a large network, we may imagine shrinking a network's width or depth as analogous to fixing certain subspaces of the network parameters (for example, reducing width is equivalent to setting many parameters to $0$). For analytical purposes, we model this process as selecting a *random* subspace of a larger network.
> > >
> > > **Interpretation of the Error Bound**
> > >
> > > Regarding the behavior of the error bound in Theorem 1, you are correct that it becomes looser when $p$ is small and $t$ is large, even if the product $pt$ remains constant. While scale-time equivalence holds more precisely as model scale grows (holding $pt$ fixed), it conversely may not hold as training time grows (holding $pt$ fixed). We are happy to clarify any parts of our paper if they gave the impression that scale time equivalence would hold $t$ goes to infinity and $pt$ is fixed.
> > >
> > > **Capacity and Scale-Time Equivalence**
> > >
> > > We understand your concern about the model's ability to fit the data when $p$ is small even given a very large training time.
> > > In our framework, assuming scale-time equivalence holds, the effective capacity of the model is determined by the product $pt$ rather than $p$ as it is conventionally. Thus, if $pt > D$ then, we will be able to fit the training data. Otherwise, we will not be able to fit the training data. Understandably, this can be counterintuitive because we usually think of $p$ as corresponding to model capacity (the amount of training data that can be fit by the model). However, at least in settings where-scale time equivalence holds, the appropriate measure of model capacity must be a function of $pt$.
> > >
> > > Please let us know if there are other concerns you would like us to address. Your feedback is invaluable in refining our work and enhancing its clarity.

---

### Official Review · Reviewer_qKW1 · 2024-11-03

**Soundness:** 2
**Presentation:** 2
**Contribution:** 3
**Rating:** 6
**Confidence:** 3

**Summary:**

This paper analyzes a simple proxy model for neural network optimization, gradient flow on a random feature model. It theoretically shows that under certain conditions, particularly on the complexity of the objective function and Lipschitz continuity of the gradients and second derivative, the speed of convergence depends on the product of the model size and time. The authors interpret this result as an equivalence of time and scale in optimization. The paper experimentally explores whether this relation holds for very small scale MLPs and conv-nets on simple vision tasks like MNIST and CIFAR. They then use time-scale equivalence to offer additional insights into double descent in terms of training time, parameter count and dataset size.

**Strengths:**

- An interesting theoretical take on scaling and double descent.
- Mostly well written, clear figures.

**Weaknesses:**

- The claims made in the paper seem too strong. The abstract mentions theoretical characterization of deep neural networks multiple times but the paper only theoretically analyzes simple proxy models. It also mentions large scale neural networks but all experiments in the paper are performed on small MNIST / SVHN / CIFAR networks. The same goes for the claims in the contribution list in the introduction.
- The paper does not sufficiently discuss its limitations (e.g. in terms of the weaknesses mentioned here).
- The neural network experiments are done on very small scale vision tasks which is typically not where neural scaling laws are valuable.
- The applicability of the random feature model analysis to neural networks is not sufficiently justified.
- Some parts of the paper are not very clear (see below).

**Edit: The authors have promised changes that I believe would sufficiently address these weaknesses. Conditioned on those changes I no longer have significant concerns and have raised my score.**

**Questions:**

S3.1: I think this section in general is not very clear and could benefit from additional explanations and better arguments for the applicability to neural networks.

L130: Neural networks with SGD often converge to flat minima when trained with a high learning rate. However, I am not sure the projection K would be fixed throughout training nor that a gradient flow model can capture the same behavior.

L139: Here you give an example of how a small neural network can be seen as a subnetwork of a larger model. This would likely imply setting various weights to zero or one for identity and so on. This would likely not be a random matrix that looks anything like R.

L152: Why can you assume that eta is not a function of p? In actual neural network training the learning rate often varies with the width and depth of the network being trained (see e.g. various muP literature).

L209: What do you mean by “minimum amount of training time required to achieve non-zero generalization”?

L244: The cube root argument should also be justified in the main body along with its applicability to neural networks. The model this is derived for seems a lot closer to the previous theoretical model than neural networks so only applying it to neural networks feels odd.

L252:  Why not tune your learning rates in general? As mentioned before, the learning rate often varies with the network width and depth. Using a fixed rate seems unprincipled.

S4: This section is very light on details. Overall the paper feels a bit crammed, trying to cover more material than can be done well on 10 pages.

---

> ### Author Response · Authors · 2024-11-22
>
> We appreciate the reviewer's thoughtful feedback and the opportunity to address the concerns raised. Below, we provide clarifications and additional details to strengthen our manuscript.
>
> **Strong Claims and Scope of the Paper**
>
> We acknowledge the reviewer's concern regarding the strength of our claims. Our intention is to present a theoretical and empirical analysis that provides insights into the relationship between model scale and training time in neural networks. While our theoretical results are derived under certain assumptions, we believe they offer valuable perspectives that are applicable beyond the specific settings studied.
>
> We have added a discussion of the limitations in the revised manuscript, specifically acknowledging that our experiments are conducted on small-scale vision tasks and that we currently lack a full theoretical understanding of the cube root scaling. We are committed to refining our claims to accurately reflect the scope and applicability of our work.
>
> **Applicability of the Random Subspace Model to Neural Network**
>
> Please refer to our general response, where we clarify the rationale behind our theoretical model. Our random subspace model is designed to capture key aspects of neural network training, including nonlinearity and overparameterization. By considering a general differentiable loss function and parameterizing the model in terms of a random subspace, we aim to abstract the effect of increasing model size in practical neural networks.
>
> **Clarity of Section 3.1**
>
> We appreciate the reviewer's request for additional explanations in Section 3.1. We refer the reviewer to our general response; if these explanations are satisfactory, we are happy to integrate them into our revised Section 3.1. We would appreciate if the reviewer could point out any specific parts that need further clarification.
>
> **Specific Line Comments**
>
> - Line 130: We acknowledge that in practice, the projection $K$ may evolve during training. In our theoretical model, we assume $K$ is fixed to simplify the analysis and focus on the relationship between model scale and training time. This assumption allows us to derive analytical results.
> - Line 139: Selecting a random subspace of parameters is conceptually similar to selecting a random subset of parameters to update. In fact, up to rotations in the parameter space, selecting a random subset and selecting a random linear subspace are equivalent.
> - Line 152: We use a fixed learning rate in our theoretical analysis to isolate the effect of model scale and training time. We agree that in practice, learning rates can vary with model size, as explored in the μP literature. However, controlling hyperparameters like learning rate is also common in much theoretical literature on scaling laws.
> - Line 209: By "minimum amount of training time required to achieve non-zero generalization," we refer to the number of epochs needed for the model to perform better than random chance on the test set.
> - Line 244: The cube root argument is detailed in Appendix B. We recognize the importance of this scaling and we are happy to summarize the key points in more detail if it would be helpful. Nevertheless, we view this as not a central contribution to the main focus of our paper.
> - Line 252: We used a fixed learning rate to maintain consistency across experiments. This approach is common in studies focusing on specific factors like model size, as it helps isolate their effects. Nevertheless, we acknowledge that tuning learning rates can impact performance.
>
> **Section 4 Details**
>
> We appreciate the feedback on Section 4 and are happy to include any specific details you suggest to improve clarity.

---

> > ### Comment · Reviewer_qKW1 · 2024-11-26
> >
> > I thank the authors for their clarifications and updates to the manuscript. I think emphasizing the theoretical nature of the work more, especially in the abstract, is important for matching it with the right type of reader / reviewer. I still have significant doubts about the applicability of the random subspace model to neural networks in practice. The main reasons remain:
> > * I strongly suspect that $K$ will generally change over time which may not only simplify the analysis but also make it inapplicable to neural networks. Specifically, I think that the statement *"This is reasonable for neural networks: it is well known
> > that neural networks trained by stochastic gradient descent tend towards flat minima of their loss landscapes, thus revealing many redundant dimensions in the network (i.e. only a low-dimensional subspace affects the network output)."* does not apply for a constant $K$ throughout training, although I agree that such a $K$ with a low effective rank may exist at the end of training. If this is not the case then I would recommend clarifying / expanding on this in the future.
> > * The use of gradient flow and constant learning rates across different network sizes is not convincing to me. I understand that this may be used in other theoretical works, but I think it should be justified further here as this is crucial to your main results (they would not hold if we could tune / modify the learning rate for each width, if I understand correctly).
> > * I think the cube root scaling is an important factor in your work because otherwise the experiments don't align well with your theoretical predictions. I still think it is strange that this is derived for a linear model similar to the original theoretical model rather than making some arguments specific to neural networks.
> >
> > Regarding *"we refer to the number of epochs needed for the model to perform better than random chance on the test set"*, I find it somewhat surprising how long this takes in some of your experiments. In my experience this almost always happens within a handful of gradient steps or at least within an epoch (given a balanced non-adversarial dataset, a well tuned learning rate, and a decent initialization).

---

> > > ### Author Response · Authors · 2024-11-30
> > >
> > > Thank you for your thoughtful feedback and for taking the time to engage deeply with our work. We appreciate your suggestions and would like to address your remaining concerns.
> > >
> > > **Applicability of the Random Subspace Model and the Constancy of K**
> > >
> > > We acknowledge that in practical neural networks, the effective directions in parameter space that significantly influence the output can change during training. Our assumption of a fixed $K$ is indeed a simplification intended to make the analysis tractable. We agree that in a real neural network, $K$ may not remain constant throughout training, and this could impact the applicability of our theoretical results. In our revised manuscript, we will clarify this assumption and discuss its implications more thoroughly. We will also explore how a time-varying $K$ could affect the scale-time equivalence and suggest it as an important direction for future research to enhance the model's practical relevance.
> > >
> > > **Use of Gradient Flow and Constant Learning Rates**
> > >
> > > We understand your reservations. Our theoretical analysis employs gradient flow and constant learning rates to provide foundational insights into the relationship between model scale and training time. We recognize that in practice, learning rates are often tuned for different network sizes to achieve optimal performance. Adjusting the learning rate can indeed influence the scale-time relationship we are investigating, and we consider it an important direction of future work.
> > >
> > > **Cube Root Scaling and Its Derivation**
> > >
> > > Regarding the cube root scaling and its derivation, we appreciate that this aspect is important for aligning our experiments with the theoretical predictions. The cube root scaling emerged from empirical observations in our experiments, where it provided consistent results across different architectures and datasets. We do note that our heuristic argument for this does take into account some aspects specific to neural networks: namely that the number of weights in a neural network is the square of the width, and the number of features is linear in width. We believe making a more refined and rigorous argument for this cube root scaling is an interesting direction for future work.
> > >
> > > **Time to Achieve Better Than Random Performance**
> > >
> > > You mentioned being surprised at how long it takes in some of our experiments for the models to perform better than random chance. This discrepancy is because in Figure 2, we plot the time to achieve better than random chance performance for *different dataset sizes.* Most of the results plotted are with datasets smaller than the original, and with smaller datasets we would expect more time required to start generalizing. In fact, with the full-sized dataset (the darkest lines in Figure 2), typically models can start generalizing right away or in less than 1 epoch.
> > >
> > > We appreciate your valuable feedback, which has helped us identify areas where further clarification is needed. We are committed to enhancing the transparency and rigor of our study, and we hope that our responses address your concerns. Please let us know if there are additional questions or suggestions you may have.

---

> > > > ### Comment · Reviewer_qKW1 · 2024-12-01
> > > >
> > > > Thank you for the clarifications. Overall I think this work presents an interesting theoretical direction and should simply be presented as such rather than a paper with immediate practical implications for neural network training. My main concerns remain the applicability to neural networks but the importance of this greatly depends on the claims made in the abstract and introduction. If the paper presented itself as a theoretical work and was more upfront about its limitations including the exact theoretical model analyzed, I would not oppose its acceptance with the other revisions the authors have promised:
> > > > * The abstract should explicitly summarize the theoretical model (random subspace model, gradient flow) and state that it is a coarse approximation of neural network training.
> > > > * The remaining theoretical claims should say something like "for this simplified model we can show x y z..." rather than claim to show this for neural networks.
> > > > * The experimental claims should be toned down a bit as well, mentioning the scale of the experiments e.g. "In our experiments we verify this for small MLPs and CNNs of different widths trained on CIFAR-10 and MNIST". I would also recommend avoiding mentioning "large models" or "practical deployment of neural networks". These claims would have to be backed by experiments on at least "small" scale transformers like ~100M parameter GPT models.
> > > > * In general being a bit more cautious in the claims i.e. "These laws explain several previously unexplained phenomena" should be at least "could explain".
> > > > * Similar changes would have to be made to the introduction.
> > > >
> > > > Regarding the experiments, I am not sure the "time to achieve better than random performance" is the right metric to measure here. Have you tried other thresholds like the time to reach 50% or 90% of final train or validation performance for example? The time to better than random performance seems a bit arbitrary and vague. For example, for a fully random initialization on a perfectly balanced test dataset, I would expect the time to obtain infinitesimally better than random predictions on average (averaged over the outputs of models with different seeds) to be one gradient step for the right learning rate. For datasets that are not perfectly balanced or instantiations that happen to get worse than random accuracy at initialization, I would expect this to only take a few steps (although this depends on the learning rate and potentially the variance used to initialize the model).

---

> > > > > ### Author Response · Authors · 2024-12-01
> > > > >
> > > > > Thank you for your thoughtful feedback and for your constructive suggestions on how to improve our manuscript. We appreciate your recognition of the theoretical contributions of our work and your guidance on how to present it more appropriately.
> > > > >
> > > > > **Framing of Paper**
> > > > >
> > > > > We understand your concern regarding the applicability of our theoretical model to practical neural network training. We agree that framing our paper as a theoretical study that provides insights into neural network scaling laws, rather than one with immediate practical implications, would more accurately reflect the nature of our work.
> > > > >
> > > > > We are happy to make all the changes you have suggested including to the abstract and introduction and our theoretical and empirical claims throughout the paper.
> > > > >
> > > > > **Choice of metric**
> > > > >
> > > > > We agree with your arguments regarding our "time to achieve better than random performance" metric and believe it deserves further investigation. Preliminarily, we found that train and test performance initially stays near random chance early in training before appearing to improve suddenly, with this point being later for smaller datasets. This type of phase transition behavior has been theoretically explained in prior literature (e.g. Saxe et al. 2013).
> > > > >
> > > > > Nevertheless, we agree that theoretically, after one gradient step, the performance should be better than random chance performance on average. To address this seeming discrepancy, we will re-analyze our experimental data using different performance thresholds, and assess under which conditions when our observed trade-off curve still holds.
> > > > >
> > > > > We are committed to revising our manuscript to address your concerns fully. By presenting our work as a theoretical study with carefully qualified claims and acknowledging its limitations, we aim to provide valuable insights while maintaining scientific rigor. Thank you again for your thoughtful and constructive comments.

---

> > > > > > ### Comment · Reviewer_qKW1 · 2024-12-01
> > > > > >
> > > > > > Thank you. With the promised revisions I no longer have significant concerns and have raised my score.

---

### Official Review · Reviewer_S73m · 2024-11-04

**Soundness:** 1
**Presentation:** 3
**Contribution:** 2
**Rating:** 3
**Confidence:** 4

**Summary:**

The paper put forward the hypothesis that increasing the training time model is in direct relationship with the model size: a small model trained for a long time is functionally equivalent to a larger model trained for a shorter time. The authors devise a simple theoretical model where this behavior can be analytically shown. They also experimentally show how this can be empirically verified in neural network training, and how their theory fits in the double descent phenomenon. The experimental results are achieved with MLPs and CNNs on computer vision benchmarks such as SVHN, CIFAR-10 and MNIST.

**Strengths:**

1. The idea of functional equivalence between models at different scales is potentially very attractive for modern practices of neural network training, where the (training) compute-optimal trade-offs have to be balanced with the potential benefits of having a smaller model at inference time.

2. The idea of formalizing the relationship between training time and scale is also interesting from a theoretical perspective. In gradient descent, training time has been linked to various forms of explicit regularization, and theoretically investigating its connection to model scale is especially interesting in the context of neural scaling laws.

**Weaknesses:**

I think that despite the captivating and well-posed questions, the paper fails to deliver on several aspects. Namely:

1. The authors claim to theoretically demonstrate that scaling the size of a neural network is equivalent to increasing its training time. The connection between the model studied here and neural networks is unclear/misleading, and some design choices are not thoroughly explained. The theoretical model has the unusual structure of equation 2, which has no connection to theoretical models for neural networks. Also, the model is linear. Also, why are $r,p,P$ all different? Why couldn’t you only have a large dimensional ambient space and a constant-sized low-rank subspace?

2. I would have appreciated it if the connection to neural networks were toned down, and the proposed model motivated in its own right. I do not understand why the authors haven't studied more established settings, where for instance the double descent phenomenon has been shown, or where scaling laws have been tractably derived. Starting from there, it would be nice to understand which (and why) additional assumptions have to be made. With this respect, why haven’t the authors approached the problem from more standard settings in neural network theory (e.g. teacher-student [1] or random feature models [2])?

3. The concept of *minimal capacity to fit the data to a certain loss level* is crucial here, but never satisfiably addressed. In fact, an underlying assumption that is never made by the authors is that the model can fit the data at any scale (only stating “Consider a large $P$ dimensional model”). Otherwise, it would not be possible for a small model to match the performances of a large model, no matter how long the small model has been trained (e.g. imagine training a model with very small width on complicated datasets).

4. The definition of **model scale** as $N^{1/3}$ ($N$ is the number of parameters) put forward by the authors is a heuristic, which is not convincing enough given its importance in the experiments on neural networks. More concretely, the authors define the model scale as "the maximum number of training points that can be fit by the network", which is set to $N^{1/3}$. The number of training points that can be fit by a model is a well-studied and hard problem. For arbitrary data points, the complexity of a function class determines how many data points can be fit. In general, it depends on the data's complexity and the model's flexibility. I find that the cube root of the number of parameters is arbitrary and ignores the data dependence. As the authors detail in the “experimental details” Section in the appendix, *“The width parameter was set to values of 1, 2, 5, 10, 20, 50, and 100. When we refer to ”model scale”, we quantify this as the cube root of the number of network parameters rather than the value of the width parameter.”* In the current form, the scale-time equivariance narrative heavily relies on the cubic root assumption, and the heuristic argument provided is insufficient considering its importance in the paper’s narrative.

5. In section 5.3, it is assumed that larger models have a smaller time to interpolation threshold. However, in the experiment of Figure 5, it appears that interpolation has not happened yet, as there are no U-shaped curves, which are the footprint of double descent. It would be nice to reproduce some of the results on double descent both in model size and training time to better substantiate some of the claims. In this sense, we would expect the interpolation threshold (peak of double descent curve) to vary with model size and training time predictably according to the scale-time equivariance theory.

6. The authors only test two learning rates, which makes it impossible to establish whether the chosen learning rate is optimal. Also, usually the optimal learning rate shifts with scale, and this is not addressed by the authors.

7. It is well-known that there is a correspondence between training time and regularization strength (e.g. [3]). The proposed theory put forward the interesting hypothesis of a similar connection between training time and model scale. I would say that mentioning this line of work would better help this paper’s positioning in the current literature.

8. In lines 367-269 a few articles “the” missing: If noise components →  if the noise components, etc..


[1] A Dynamical Model of Neural Scaling Laws

[2] A Solvable Model of Neural Scaling Laws

[3] A Continuous-Time View of Early Stopping for Least Squares Regression (https://proceedings.mlr.press/v89/ali19a)

**Questions:**

See weaknesses.

---

> ### Author Response · Authors · 2024-11-22
>
> We thank the reviewer for their thoughtful comments and for highlighting areas where our manuscript could be improved. We address the specific concerns below.
>
> **Concerns on Theoretical Setup**
>
> Please see our general response above, where we clarify the theoretical model and its applicability to neural networks. Our model is designed to capture essential characteristics of neural network training, including nonlinearity and overparameterization. By abstracting the network's output dependency on certain parameter directions through $\alpha = K\beta$, we aim to provide a general framework that is relevant to deep neural networks.
>
> **Minimal Capacity to Fit the Data**
>
> The reviewer raises an important point regarding the model's capacity to fit the data. In our theoretical result (Theorem 1), the error bound contains a term inversely proportional to the square root of the model scale $p$ (i.e., $\frac{1}{\sqrt{p}}$​). This indicates that the scale-time equivalence holds more precisely for sufficiently large models. As $p$ becomes small, the error bound becomes looser, and the equivalence may break down.
>
> We emphasize that our analysis does not assume that the model can fit the data at any scale; we do not assume that models are large enough to interpolate the training data. In fact, the maximum model size at which scale-time equivalence breaks down may be completely unrelated to its ability to fit the training data.
>
> **Cube Root Relationship Between Parameters and Model Scale**
>
> The reviewer correctly points out that our definition of model scale as the cube root of the number of parameters is heuristic. We based this choice on empirical observations where the cube root scaling provided consistent results across different architectures and datasets in our experiments.
>
> This scaling reflects the idea that the effective capacity of neural networks increases with both width and depth, and their combined effect on parameter count is nonlinear. We agree that the relationship between parameters and model scale can vary depending on the architecture and data; the cube root scaling is just what we observed with our standard architectures on standard vision benchmarks.
>
> **Concerns on Figure 5**
>
> Regarding the observations in Figure 5, we note that in our experiments, both test and train error decrease monotonically with the amount of data, even as model size varies. This behavior aligns with prior work, such as Nakkiran et al. (2019), where increasing data volume generally leads to improved performance on CIFAR-10. Thus, we actually do not expect to see double-descent like behavior on this plot, consistent with the finding that it does not always appear.
>
> **Comments on Learning Rate**
>
> In our experiments, we used a fixed learning rate across different model scales to maintain consistency and to focus on the effects of model size and training time.
>
> Our theoretical analysis assumes a constant learning rate, and varying it would introduce additional variables that could confound the relationship we are studying. We acknowledge that in practice, adjusting the learning rate with model size can be beneficial. However, our findings on scale-time equivalence are based on the assumption of a fixed learning rate, which is a common practice in theoretical studies to isolate specific effects.
>
> **Related Work**
>
> We appreciate the references to additional related work, including the papers on training time as a form of regularization. We have included these references in the revised manuscript.
>
> **Typos**
>
> Thank you for pointing out the missing articles in lines 267-269. We have corrected these typos in the revised manuscript.

---

> ### Comment · Reviewer_S73m · 2024-11-24
> **Response to Rebuttal**
>
> I sincerely thank the authors for the rebuttal. Indeed, a few concerns (e.g. the learning rate issue) are resolved
>
> I think the main concern with the connection of the proposed model and neural networks, shared with other reviewers in many forms, still persists. In fact, I find it a bit vague that the random subspace is a correct abstraction of the model's depth or width, in the sense these quantities introduce fundamental challenges to optimization and generalization in neural networks. Also, my questions regarding the comment *[...] the model is linear. Also, why are $r, p, P$ all different? Why couldn’t you only have a large dimensional ambient space and a constant-sized low-rank subspace?* still persist. In my view, I would suggest better justifying the model, and potentially considering studying a random feature model is very close and better tied to the literature. In the current form, I believe that the proposed model and heuristics on the relationship between model scale and the number of parameters are used to simplify the problem, rather than finding good abstraction to model neural networks. Also, as they are not introduced earlier in the literature, they should be more thoroughly justified.

---

> > ### Author Response · Authors · 2024-11-25
> >
> > Thank you for your engagement with our work and for acknowledging that some of your concerns have been resolved. We appreciate the opportunity to address your remaining questions regarding the connection between our theoretical model and neural networks.
> >
> > We understand that the abstraction of the random subspace as a representation of model width or depth may seem vague, and we apologize for not providing sufficient justification in our initial manuscript. Our intention was to develop a theoretical model that captures the essence of how increasing model size (through width or depth) affects training dynamics in neural networks, specifically in the context of scale-time equivalence.
> >
> > **Why Use a Random Subspace Model?**
> >
> > The assumption that we control a random $p$-dimensional subspace of a larger set of parameters $\beta$ reflects the practical scenario of increasing or decreasing a neural network's width or depth. Starting from a large network, we may imagine shrinking a network's width or depth as analogous to fixing certain subspaces of the network parameters (for example, reducing width is equivalent to setting many parameters to $0$). For analytical purposes, we model this process as selecting a *random* subspace of a larger network. We clarify that this model is not uniquely tied to neural networks; it could model scale changes in other similarly flexible model classes as well.
> >
> > While our model shares similarities with random feature models, which are inherently linear in their parameters, our setup does not assume linearity. The loss can be an arbitrarily nonlinear function of the model parameters, allowing for greater generality than random feature models.
> >
> > **Addressing the Differences Between $P$, $p$, and $r$**
> >
> > We recognize that the use of three different dimensions ($P$, $p$, and $r$) may seem unnecessarily complex. Here's why each is important in our model:
> >
> > - $P$: The total number of parameters in the large ambient space. This represents the full parameter space of a potentially very large neural network.
> > - $p$: The dimensionality of the subspace we control. By varying $p$, we model the effect of changing the model size (e.g., adjusting the width or depth of the network).
> > - $r$: The dimensionality of the effective parameter space that influences the output through $\alpha = K\beta$. This reflects the idea that, due to parameter redundancy and the network's architecture, the function implemented by the network depends on a lower-dimensional combination of parameters.
> >
> > Having these distinct dimensions allows us to consider the effect of changing model size $p$ while not changing the underlying dimensionality of the task $r$. While we appreciate the reviewer's suggestion, we unfortunately cannot model the effects of changing model size by only including $P$ and $r$.
> >
> > **Why Not Use a Random Feature Model?**
> >
> > Random feature models are a valuable tool in theoretical analyses, and there have been many theoretical analyses of scaling laws with random feature models. However, random feature models are still fundamentally linear models: they do not capture the nonlinear effects that parameters can have on model outputs.
> >
> > Our goal was to develop a model that, while abstract, allows for nonlinearity the model and captures the adaptive nature of learning in neural networks. The random subspace model achieves this by not making assumptions about the specific form of $L(\alpha)$ and by allowing the parameters to influence the output through a projection $K$ that can represent complex dependencies.
> >
> > We appreciate your suggestion and understand the importance of grounding our theoretical model more firmly in established literature. Our intention is not to oversimplify the problem but to provide a framework that captures essential aspects of neural network training while allowing for new analytical insights into the scale-time trade-off.
> >
> > Please let us know if there are additional questions or if there are specific aspects you believe we should elaborate on further. Your feedback is invaluable in helping us improve our work.

---

### Author Response · Authors · 2024-11-22
**General Response**

We thank the reviewers for their careful reading of our manuscript and for their constructive feedback. We appreciate the opportunity to clarify the key aspects of our work, particularly regarding our theoretical model and its applicability to neural networks.

**Clarifying the Theoretical Model and Its Applicability**

A central concern raised by the reviewers is the perceived disconnect between our theoretical model and neural networks, with some suggesting that our model is limited to linear settings or lacks relevance to neural networks.

We would like to clarify that our theoretical model is designed to capture essential characteristics of neural network training, including nonlinearity and possible overparameterization. Specifically:

- General Nonlinear Loss Function: In our model, the loss function $L(\alpha)$ can be any differentiable function, potentially non-convex and arising from a deep neural network. The parameters $\alpha = K\beta$ represent the function implemented by the network, where $K$ is a fixed projection matrix. This setup is intended to abstract the idea that the network's output depends on certain directions in parameter space.

- Random Subspace of Parameters: The assumption that we control a random $p$-dimensional subspace of the parameters reflects the practical scenario where we increase the width or depth of a neural network. In neural networks, expanding the architecture introduces new parameters in random directions (due to random initialization). Thus, our model captures the effect of increasing model size on training dynamics.

- Scale-Time Equivalence: Our main theoretical result demonstrates that scaling the size of the parameter subspace ($p$) is equivalent to increasing training time ($t$), under gradient flow dynamics. This result holds for general differentiable loss functions and does not rely on linearity. It reveals a fundamental trade-off between model size and training time that is applicable to neural networks trained with gradient-based methods.

**Why Not Use Standard Theoretical Models?**

Some reviewers suggested using more standard theoretical models, such as random feature models or teacher-student setups. While these models are valuable for certain analyses, we deliberately chose a different approach for the following reasons:

- Generality: Our model is intended to be more general and less reliant on specific architectural assumptions. For instance, random feature models are linear models in the parameters, thus not capturing the nonlinearities of neural networks. Teacher-student setups often also operate under the assumption of a particular architecture. By not restricting ourselves to particular models, we aim to derive insights that are broadly applicable across different neural network architectures.

- Focus on Scale-Time Trade-off: Traditional models may not readily capture the specific relationship between model scale and training time that we are investigating. Our model is specifically designed to analyze this trade-off in a principled way.

We appreciate the reviewers' concerns and believe that they stem from misunderstandings that we can address by clarifying our theoretical model and its relevance to neural networks. Our work provides a novel perspective on the interplay between model scale and training time, offering insights that are both theoretically sound and practically relevant.

We are happy to further clarify any of these points as we believe they are central to understanding our submission.

---

### Meta-Review · Area_Chair_nWyC · 2024-12-20

**Metareview:**

This paper explores the concept of "scale-time equivalence," suggesting that smaller neural networks trained for extended periods can match the performance of larger models trained briefly, supported by a theoretical linear model and experiments on small-scale vision tasks. The work offers a potentially valuable perspective on training efficiency and suggests a novel explanation for double descent, improving our understanding of model scaling. However, the theoretical analysis is limited to a simplistic linear model that is weakly connected to real neural networks, the experimental validation relies on small-scale models and datasets, and the core assumption of scale-time equivalence hinges on a questionable definition of model scale, ultimately raising concerns about the paper's generalizability and practical applicability. I do not recommend acceptance at this time.

**Additional Comments On Reviewer Discussion:**

Reviewers raise concerns about the applicability of the authors' simplified theoretical model to real neural networks, particularly regarding the model's linearity, the use of a fixed random subspace to represent changes in model size, and the reliance on a heuristic cube-root relationship between model parameters and scale.  The authors defend their model as a general abstraction capturing essential aspects of neural network training, emphasizing its nonlinearity and the focus on the scale-time trade-off, but acknowledge limitations in experimental scale and the need for further theoretical grounding, ultimately positioning their work as a primarily theoretical contribution with insights into scaling laws rather than a prescription for immediate practical application. They further address concerns about specific experimental details and metric choices while committing to revising the manuscript to clarify their model, assumptions, and limitations, along with toning down claims of direct applicability to large-scale neural network training. Ultimately, the rebuttal did not change the opinion of most reviewers, and the sentiment remains that this paper is not suitable for publication at this time.

---

### Decision · Program_Chairs · 2025-01-22

Reject